



# Sequential assimilation of satellite-derived vegetation and soil moisture products using SURFEX_v8.0: LDAS-Monde assessment over the Euro-Mediterranean area

Clément Albergel[1,*], Simon Munier[1] , Delphine Jennifer Leroux[1], Hélène Dewaele[1], David Fairbairn[1,2], Alina Lavinia Barbu[1], Emiliano Gelati[1,3], Wouter Dorigo[4], Stéphanie Faroux[1], Catherine

Meurey[1], Patrick Le Moigne[1], Bertrand Decharme[1], Jean-Francois Mahfouf[1], Jean-Christophe Calvet[1]

[1] CNRM UMR 3589, Météo-France/CNRS, Toulouse, France

[2] Now at Imperial College, London, UK

[3] Now at Joint Research Centre, European Commission, Ispra, Italy

[4] Department of Geodesy and Geo-Information, TU Wien (Vienna University of Technology), Vienna, Austria

[*] Corresponding author, Clement Albergel: clement.albergel@meteo.fr

**Abstract-** In this study, a global Land Data Assimilation system (LDAS-Monde) is tested over Europe and the Mediterranean basin to increase monitoring accuracy for land surface variables. LDAS-Monde is able to ingest information from satellite-derived surface Soil Moisture (SM) and

Leaf Area Index (LAI) observations to constrain the Interactions between Soil, Biosphere, and Atmosphere (ISBA) land surface model (LSM) coupled with the CNRM (Centre National de Recherches Météorologiques) version of the Total Runoff Integrating Pathways (ISBA-CTRIP) continental hydrological system. It makes use of the $CO_2$-responsive version of ISBA which models leaf-scale physiological processes and plant growth. Transfer of water and heat in the soil rely on a

multilayer diffusion scheme. Surface SM and LAI observations are assimilated using a simplified extended Kalman filter (SEKF), which uses finite differences from perturbed simulations to generate flow-dependence between the observations and the model control variables. The latter include LAI and seven layers of soil (from 1 cm to 100 cm depth). A sensitivity test of the Jacobians over 2000-2012 exhibits effects related to both depth and season. It also suggests that observations

of both LAI and surface SM have an impact on the different control variables. From the assimilation of surface SM, the LDAS is more effective in modifying soil-moisture from the top layers of soil as model sensitivity to surface SM decreases with depth and has almost no impact from 60 cm downwards. From the assimilation of LAI, a strong impact on LAI itself is found. The LAI assimilation impact is more pronounced in SM layers that contain the highest fraction of roots (from 10 cm to 60 cm). The assimilation is more efficient in summer and autumn than in winter and





spring. Assimilation impact shows that the LDAS works well constraining the model to the observations and that stronger corrections are applied to LAI than to SM. The assimilation impact's evaluation is successfully carried out using (i) agricultural statistics over France, (ii) river discharge observations, (iii) satellite-derived estimates of land evapotranspiration from the Global Land Evaporation Amsterdam Model (GLEAM) project and (iv) spatially gridded observations based estimates of up-scaled gross primary production and evapotranspiration from the FLUXNET network. Comparisons with those four datasets highlight neutral to highly positive improvement.

## 1 Introduction

Land surface models (LSMs) forced by gridded atmospheric variables and their coupling with river routing models are important for understanding the terrestrial water and vegetation cycles (Dirmeyer et al., 2006). These LSMs need to simulate biogeophysical variables like surface and root zone soil moisture (SSM and RZSM, respectively), Leaf Area Index (LAI) in a way that is fully consistent with the representation of surface/energy flux and river discharge simulations. Soil Moisture (SM) is an essential component in partitioning incoming water and energy over land, thus affecting the variability of evapotranspiration, runoff and energy fluxes (Mohr et al., 2000). By controlling land surface temperature and plant water stress, evapotranspiration and infiltration of precipitation, soil moisture drives ecosystem dynamics, biodiversity and food production, regulates $CO_2$ emissions (uptake) by the land surface and impacts natural hazards such as floods and droughts (Seneviratne et al., 2010). The role of soil moisture as a regulator for various processes in the terrestrial ecosystem such as plant phenology, photosynthesis, biomass allocation, soil respiration, hence the terrestrial carbon balance, has also clearly been established (Ciais et al., 2005; Van der Molen et al., 2012; Carvalhais et al., 2014; Reichstein et al., 2013). The seasonal dynamics of vegetation properties, like LAI, are connected to soil moisture dynamics (Kochendorfer and Ramirez, 2010). Both the simulation of hydrological processes and the exchange of water vapour and $CO_2$ between the vegetation canopy and atmosphere interface are strongly influenced by LAI (Jarlan et al., 2008; Szczypta et al., 2014).

Global observations of land surface variables are now operationally available from spaceborne instruments and they can be used to constrain LSMs through Data Assimilation (DA) techniques as demonstrated by several authors (e.g., Reichle et al., 2002; Draper et al., 2011, 2012; Dharssi et al., 2011; Barbu et al., 2011; de Rosnay et al., 2013, 2014; Barbu et al., 2014; Boussetta et al., 2015;



Fairbain et al., 2017). Recent studies (e.g., Traore et al., 2014) have demonstrated that a model that best performs for soil moisture does not necessarily best perform for plant productivity, highlighting the need to jointly use soil moisture and vegetation observations to improve global and continental eco-hydrological/carbon cycle models (Wang et al., 2012; Kaminski et al., 2013). Several studies
demonstrated the benefit of jointly assimilating SSM and LAI on the representation of RZSM (e.g., Sabater et al., 2008) and $CO_2$ flux (e.g., Albergel et al., 2010, Barbu et al., 2011).

Within the SURFEX modelling system (SURFace EXternalisée, Masson et al. 2013) the $CO_2$-responsive version of ISBA (Interaction between Soil Biosphere and Atmosphere) LSM (Noilhan and Mahfouf, 1996; Calvet et al., 1998, 2004; Gibelin et al., 2006) allows the representation of various
land surface processes, including evapotranspiration and SM evolution. It is also capable of modelling photosynthesis and vegetation growth. The evolution of the simulated LAI and vegetation biomass changes in response to the meteorological forcing conditions. In previous studies, Barbu et al. (2014), Fairbairn et al. (2017) tested a combined assimilation of SSM and LAI in this $CO_2$ responsive version of ISBA over France within SURFEX. They used the force-restore version of
ISBA (with three layers of soil), a Simplified formulation of an Extended Kalman Filter (SEKF) with a 24-h assimilation window and hourly meteorological forcing from the SAFRAN reanalysis (Quintana-Seguı et al., 2008; Habets et al., 2008) at 8km scale. Fairbairn et al. (2017), also made a posterior offline use of runoff and drainage fields from ISBA to run the MODCOU hydrological model (Habets et al., 2008) to evaluate the added value of the joint assimilation of LAI and SSM on
the representation of river discharge over France. However, the assimilation was not successful in improving the representation of river discharge within MODCOU compared to an open-loop (i.e. no assimilation) simulation. Following their work, the present study tests the assimilation of both satellite derived SSM and LAI at the continental scale. Further steps are made by:

- Using the most recent SURFEX_v8.0 Offline Data Assimilation implementation,

- Considering a much larger domain, Europe and the Mediterranean basin as well as a longer time period; 2000-2012,

- Using the multi-layer soil diffusion scheme of ISBA developed by Decharme et al. (2011).





- Assimilating a long term, global scale, multi-sensor satellite-derived surface soil moisture dataset (ESA CCI SSM, Liu et al., 2011, 2012; Dorigo et al., 2015, 2017) along with satellite-derived LAI (GEOV1, http://land.copernicus.eu/global/),

- Using the modified version of WFDEI observation-based atmospheric forcing dataset (Weedon et al., 2011, 2014) from the eartH2Observe project (Schellekens et al., 2017),

- Having a daily interactive coupling between ISBA and the CNRM (Centre National de Recherches Météorologiques) version of the TRIP (Total Runoff Integrating Pathways, Oki et al., 1998) river routing model (CTRIP hereafter) to simulate hydrological variables such as the river flow (Decharme et al. 2010).

Section 2 presents the LDAS-Monde system system, i.e. (i) the $CO_2$ responsive version of the ISBA LSM and the soil diffusion scheme, (ii) the CTRIP hydrological model and its coupling with ISBA, (iii) the atmospheric forcing used to drive the system, (iv) the equations of the SEKF and (v) the assimilated remotely sensed observations dataset as well as the datasets used to evaluate the analysis impact (agricultural statistics over France, river discharge, satellite-derived estimates of land transpiration and spatially gridded estimates of up-scaled gross primary production from the FLUXNET network). Section 3 investigates and discusses the model sensitivity to the assimilated observations and provides a set of statistical diagnostics to assess and evaluate the analysis impact. Finally section 4 provides perspective and future research directions.

## 2 Materials and Method

### 2.1 SURFEX offline data assimilation

The SURFEX modelling system includes the ISBA land surface model (Noilhan and Mahfouf, 1996) to calculate the soil/vegetation/snow energy and water budgets and is coupled to the TRIP (Total Runoff Integrating Pathways, Oki et al., 1998) river routing model in order to simulate the streamflow (SURFEX-CTRIP hereafter). SURFEX offline data assimilation implementation is used to set up a Land Data Assimilation System (LDAS) over Europe and the Mediterranean basin. It is defined as an offline sequential data assimilation system based on the ISBA LSM. It is capable of ingesting information from various satellite-derived observations to analyse and update SM and LAI simulated by ISBA. Analysis of ISBA prognostic variables then have an impact on the CTRIP variables (e.g., river discharge) through an interactive daily coupling (Voldoire et al. 2017). The



system is driven by WFDEI (*WATCH-Forcing*-Data-ERA-Interim) observations based atmospheric forcing dataset (Weedon et al., 2011, 2014). The main components of the LDAS (LSM, river routing system, analysis scheme and atmospheric forcing) are detailed hereafter.

120          2.1.1     *ISBA Land Surface Model,*

ISBA models the basic land surface physics requiring only a small number of model parameters. They depend on the soil and vegetation types. This study uses of the $CO_2$-responsive version of ISBA which is able to simulate the interaction between water and carbon cycles, photosynthesis and its coupling to stomatal conductance (Calvet et al., 1998, 2004; Gibelin et al., 2006). The $CO_2$-

responsive version of ISBA has been developed to allow for different biomass reservoirs for the simulation of photosynthesis and the vegetation growth. The dynamic evolution of the vegetation biomass and LAI variables is driven by photosynthesis in response to atmospheric and climate conditions. Photosynthesis enables vegetation growth resulting from the net assimilation of $CO_2$. During the growing phase, enhanced photosynthesis corresponds to a net assimilation of $CO_2$, which

results in vegetation growth from the LAI minimum threshold (prescribed as 1 $m^2m^{-2}$ for coniferous forest or 0.3 $m^2m^{-2}$ for other vegetation types). In contrast, a deficit of photosynthesis leads higher mortality rates. The total evaporative flux represents the combination of the evaporation of (i) plant transpiration, (ii) liquid water intercepted by leaves, (iii) liquid water contained in top soil layers, and (iv) the sublimation of the snow and soil ice. The $CO_2$ uptake from photosynthesis is defined as

the gross primary production (GPP) and the release of $CO_2$ is called the ecosystem respiration (RECO). The Net ecosystem $CO_2$ exchange (NEE) measures the difference between these two quantities.

ISBA has an explicit snow scheme (with 12 layers) as detailed in Bonne and Etchevers (2001) and Decharme et al. (2016). The multi-layer soil diffusion scheme version is based on the mixed form of

the Richard's equation and explicitly solves the one-dimensional Fourier law. Additionally, it incorporates soil freezing processes developed by Boone et al. (2000) and Decharme et al. (2013). The total soil profile is vertically discretised and the temperature and the moisture of each layer are computed according to the textural and hydrological characteristics. The Brookes and Corey model determines the closed-form equations between the soil moisture and the soil hydrodynamic

parameters, including the hydraulic conductivity and the soil matrix potential (Decharme et al. 2013). A discretization with 14 layers over 12m depth is used. The lower boundary of each layer is:





0.01, 0.04, 0.1, 0.2, 0.4, 0.6, 0.8, 1, 1.5, 2, 3, 5, 8 and 12 m depth (see figure 1 of Decharme et. al., 2011). The amount of clay, sand and organic carbon present in the soil are determined by thermal and hydrodynamic soil properties (Decharme et al., 2016) and are taken from the Harmonised
World Soil Database ( HWSD, Wieder et al., 2014). As for hydrology, the infiltration, surface evaporation and total runoff are accounted for in the soil water balance. The discrepancy between the surface runoff and the throughfall rate is defined by the infiltration rate.

The throughfall rate is defined as the sum of rainfall that is not intercepted by the canopy, dripping from the canopy (interception reservoir) and snow melt water. Evaporation only affects the
superficial layer, which represents the top 1 cm of soil. The soil evaporation is proportional to the relative humidity of the superficial layer. Transpiration water from the root zone (the region where the roots are asymptotically distributed) following the equations in Jackson et al. (1996). More information on the root density profile is available in Canal et al. (2014). ISBA total runoff has two contributions: the surface runoff (the lateral subsurface flow in the topsoil) and a free drainage
condition at the bottom layer. A basic TOPMODEL approach is used to compute the Dunne runoff and lateral subsurface flow from a subgrid distribution of the topography. The Horton runoff is estimated from the maximum soil infiltration capacity and a subgrid exponential distribution of the rainfall intensity.

### 2.1.2    CTRIP river routing

The present CTRIP version consists of a global streamflow network at 0.5° spatial resolution. The CTRIP model is driven by the three prognostic equations corresponding to the groundwater, the surface stream water and the seasonal floodplains. Streamflow velocity is computed using the Manning's formula (Decharme et al., 2010). The floodplain reservoir fills when the river water level overtops the riverbank and empties again when the water level drops below this threshold
(Decharme et al., 2012). Flooding impacts the ISBA soil hydrology through infiltration. It also influences the overlying atmosphere via free surface water evaporation and precipitation interception.

At last, the groundwater scheme (Vergnes and Decharme, 2012) is based on the two-dimensional groundwater flow equation for the piezometric head. Its coupling with ISBA permits accounting for
the presence of a water table under the soil moisture column allowing upward capillary fluxes into the soil (Vergnes et al., 2014). CTRIP is coupled to ISBA through OASIS-MCT (Voldoire et al.





2017). Once a day, ISBA provides CTRIP with updates on runoff, drainage, groundwater and floodplain recharges, CTRIP returns to ISBA the water table depth/rise, floodplain fraction, flood potential infiltration.

### 2.1.3    *Extended Kalman Filter*

This section describes the analysis update of the Extended Kalman Filter while its application setup is described in section 2.3.

The analysis update equation of the Extended Kalman Filter is:

$$x_a(t_i) = x_f(t_i) + K_i\left(y_o(t_i) - h_i[x_f]\right) \tag{1}$$

The "a", "f" and "o" subscripts stand for analysis, forecast and observation, respectively. $x$ is the control vector of dimension $N_x$, computed at time $t_i$, that represents the prognostic equations of the LSM $M$.

$y_o$ is the observation vector of dimension $N_y$. The Kalman gain matrix $K_i$ is computed at time $t_i$ as:

$$K_i = B H^T\left(HBH^T + R\right)^{-1} \tag{2}$$

A non-linear observation operator $h$, enables the extraction of the model counterpart of the observations:

$$y(t_i) = h(x) \tag{3}$$

$B$ and $R$ are error covariance matrices characterising the forecast and observations vectors. The cross-correlated terms represent covariances. The operator $H$ (and its transpose $H^T$) from Eq.2 is the Jacobian matrix: the linearized version of the observation operator (defined as $N_y$ rows and $N_x$ columns) that transforms the model states into the observations space. A numerical estimation of each Jacobian element is calculated by finite differences, by perturbing each component $x_j$ of the control vector $x$ by a specific amount $\delta x_j$ resulting in a column of the matrix $H$ for each integration *m*:

$$H_{mj} = \frac{\partial y_m}{\partial x_j} \approx \frac{y_m(x + \delta x_j) - y_m}{\delta x_j} \tag{4}$$

The control vector evolution from time $t_i$ to time $t_{i+1}$ is then controlled by the following equation:



$$x_f(t_{i+1}) = M_i[x_a(t_i)] \tag{5}$$

In line with previous studies (e.g, Mahfouf et al., 2009; Albergel et al., 2010; Barbu et al., 2011; de Rosnay et al., 2013; Barbu et al., 2014; Fairbairn et al., 2015, 2017) a fixed estimate of the back­ground-error variances and zero covariances at the start of each cycle are used leading to a Simpli­fied version of the Extended Kalman Filter (SEKF hereafter). The initial state at the start of a 24-hour assimilation window is analysed by assimilating the observations available over the previous 24-hour assimilation window. This approach is similar to the "simplified 2-D-Var (2-dimensional variational data assimilation scheme)" proposed by Balsamo et al. (2004) but the increments are ap­plied at the final timestep of the 24-hour assimilation window. Draper et al. (2009) found that the SEKF could generate flow-dependence from the 24-hour assimilation window and cycling the back­ground-error covariance (as in the EKF) gave no additional benefit.

## 2.2 Data and data processing

### 2.2.1 WFDEI observations based atmospheric forcing dataset

Atmospheric forcing from the WFDEI dataset (Weedon et al., 2011, 2014) is used to drive the LDAS. It spans the period 1979-2012 and contains three-hourly time intervals of: wind speed, atmospheric pressure, air temperature, air humidity, incoming shortwave and longwave radiations and solid and liquid precipitation. WFDEI originates from the ECMWF ERA-Interim reanalysis (Dee et al., 2011) with a spatial resolution of 0.5°, and is corrected with the CRU dataset (Climatic Research Unit, Harris et al., 2014) using a sequential elevation correction of surface meteorological variables plus monthly bias correction from gridded observations (e.g., precipitation data from the Global Precipitation Climatology Centre; GPCC). A more exhaustive description of the dataset is available in Schellekens et al. (2017).

### 2.2.2 ESA CCI surface soil moisture

This study makes use of a multi-sensor, long-term and global satellite-derived surface soil moisture dataset (Liu et al., 2011, 2012; Wagner et al., 2012 ; Dorigo et al., 2015) developed within The European Space Agency Water Cycle Multi-mission Observation Strategy (ESA-WACMOS) project and Climate Change Initiative (CCI, http://www.esa-soilmoisture-cci.org). Several authors (e.g., Albergel et al., 2013a, 2013b; Dorigo et al., 2015) have highlighted the quality and stability over time of the product. Despite some limitations, this data set has shown potential for assessing model





performance (Szczypta et al., 2014; van der Schrier, et al., 2013), for investigating the connection between soil moisture and atmosphere–ocean oscillations (Bauer-Marschallinger et al., 2013) as well as vegetation dynamics (Barichivich et al.,2014; Muñoz et al., 2014). This study uses the ESA

CCI SM COMBINED latest version of the product (v03.2) which merges SM observations from seven microwave radiometers (SMMR, SSM/I, TMI, ASMR-E, WindSat, AMSR2, SMOS) and four scatterometers (ERS-1/2 AMI and MetOp-A/B ASCAT) into a harmonies dataset covering the period November 1978 to December 2015. For a more comprehensive overview of the ESA CCI SM see Dorigo et al, 2015, 2017 (under review for RSE).

To assimilate SM data, it is important to rescale the observations such that they are consistent with the model climatology (Reichle and Koster, 2004; Drusch et al., 2005). The climatology of the SM data set is defined by the specific mean value, variability and dynamical range. The ISBA model climatology for each gridpoint is dependent on the dynamical range, which is calculated from the wilting point and field capacity parameters (functions of soil texture types). It is necessary to

transform the ESA CCI SSM product into model equivalent SSM to address possible mis-specification of physiographic parameters, such as the wilting point and the field capacity. The linear rescaling approach described in Scipal et al., 2008 (using the first two moments of the Cumulative Distribution Function, CDF) has been used in this study; it is a linear rescaling that enables a correction of the differences in the mean and variance of the distribution. The first two

moments, the intercept $a$ and the slope $b$ are:

$$a = \overline{SSM_m} - b \times \overline{SSM_o} \qquad (6)$$

$$b = \frac{\sigma_m}{\sigma_o} \qquad (7)$$

Where $\overline{SSM_m}$ ( $\sigma_m$ ) and $\overline{SSM_o}$ ( $\sigma_0$ ) correspond to the model and observation means (standard deviations), respectively. Barbu et al., 2014 and Draper et al., 2011 discussed the

importance of allowing for seasonal variability in the CDF matching. $a$ and $b$ parameters vary spatially and were derived on a monthly basis by using a three-month moving window over 2000 to 2012 after screening for presence of ice and urban areas. The ESA CCI SSM observations are interpolated by an arithmetic average to the $0.5°$ model gridpoints.

### 2.2.3    GEOV1 Leaf Area Index



The GEOV1 LAI is produced by the European Copernicus Global Land Service project (http://land.copernicus.eu/global/). The LAI observations are retrieved from the SPOT-VGT and PROBA-V (from 1999 to present) satellite data according to the methodology discussed in Baret et al. (2013). Following Barbu et al. (2014), the 1 km resolution observations are interpolated by an arithmetic average to the 0.5° model gridpoints, as long as 50% of the observation gridpoints are

observed (half the maximum amount). LAI observations have a temporal frequency of 10 days. Both SSM and LAI observed data set are illustrated in Figure 1 presenting averaged values over 2000-2012. Figure 1 also illustrates the studied domain.

### 2.2.4 Evaluation data sets and strategies

A common diagnostic in data assimilation is to compute (1) differences between the assimilated

observations and the model background, called the innovations, and (2) differences between the assimilated observations and the analysis, called the residuals (Barbu et al., 2011). Assuming that the system is working well, residuals have to be reduced compared to the innovations.

After evaluating innovations and residuals of SSM and LAI, analysis impact is assessed using (1) agricultural statistics over France, (2) observed river discharge over Europe, (3) satellite-derived

estimates of terrestrial evapotranspiration from the Global Land Evaporation Amsterdam Model (GLEAM, Martens et al., 2016, in review) and (4) spatially gridded estimates of up-scaled Gross Primary Production (GPP) and evapotranspiration from the FLUXNET network (Jung et al., 2009, 2011).

Smith et al. (2010a, b) demonstrated that crop simulations can be validated by agricultural statistics

on a country scale. With a finer spatial scale over France, Calvet et al. (2012) benchmarked several configurations of the ISBA LSM using agricultural statistics (Agreste, 2016), namely the correlation between yield time series and above-ground biomass ($B_{ag}$) simulations. In ISBA, $B_{ag}$ of herbaceous vegetation is made up of two components: the active biomass and the structural biomass. The former describes the photosynthetically active leaves and is linked to $B_{ag}$ by a nitrogen dilution

allometric logarithmic law (Calvet and Soussana, 2001). Calvet et al. (2012), found that $B_{ag}$ simulated by the model is in agreement with the agricultural statistics, and therefore can be used to benchmark model/system development. Yearly statistical surveys over France are provided by the Agreste portal (http://agreste.agriculture.gouv.fr/). This has enabled a database of annual straw cereal grain yield (GY) values to be established. The GY estimates are available according to



administrative unit (département) and per crop type. Following Calvet et al. (2012), Canal et al. (2014) and Dewaele et al. (2017), the GY values for rainfed straw cereals over 45 départements are used, which include barley, oat, rye, triticale and wheat. Simulated and analysed annual maximum of $B_{ag}$ are compared to GY estimates following the methodology from Dewaele et al. (2017). Although SURFEX does not directly represent GY, it is assumed that the regional-scale simulations

of above-ground biomass from a generic LSM can provide the inter-annual variability as a proxy for GY (Calvet et al., 2012; Canal et al., 2014).

Over 2000-2010, simulated and analysed discharge are compared to gauging measurements from the Global Runoff Data Center (GRDC; http://grdc.sr.unh.edu/index.html) and the Banque Hydro (http://www.hydro.eaufrance.fr/) at a monthly time step. Data are chosen over the domain presented

in Figure 1 for sub-basins with large drainage areas ($10000 km^2$ or greater) and with a long observation time series (4 years or more). It is common to express observed and simulated river discharge (Q) data in $m^3 s^{-1}$. However, given that the observed drainage areas may differ slightly from the simulated ones, scaled Q-values in $mm.d^{-1}$ (the ratio of Q to the drainage area) are used in this study. Stations with drainage areas differing by more than 15% from the simulated (analysed)

ones are also discarded. This leads to 83 stations. Impact on Q is evaluated using correlation, RMSD as well as the efficiency score ( $Eff$ ) (Nash and Sutcliff, 1970). $Eff$ evaluates the model ability to represent the monthly discharge dynamics and is given by:

$$Eff = 1 - \frac{\sum_{t=1}^{T}\left(Q_s^t - Q_o^t\right)^2}{\sum_{t=1}^{T}\left(Q_o^t - \overline{Q_o^t}\right)^2} \tag{8}$$

where $Q_s^t$ is the simulated river discharge (or analysed) at time t and $Q_o^t$ is observed river

discharge at time t. The $Eff$. can vary between $-\infty$ and 1. A value of 1 corresponds to identical model predictions and observed data. A value of 0 implies that the model predictions have the same accuracy as the the mean of the observed data. Negative values indicate that the observed mean is a more accurate predictor than the model simulation.

The GLEAM product uses a set of algorithms to estimate terrestrial evaporation and root-zone SM

from satellite data (Miralles et al., 2011). It is a useful validation tool given that such quantities are difficult to measure directly at large scales. The global evaporation model in GLEAM is mainly





driven by microwave remote sensing observations, while potential evaporation rates are constrained by satellite derived SM data. It is a well-established dataset that has been widely used to study trends and spatial variability in the hydrological cycle (e.g., Jasechko et al., 2013; Greve et al.,
2014; Miralles et al., 2014a; Zhang et al., 2016) and land–atmosphere feedbacks (e.g., Miralles et al., 2014b; Guillod et al., 2015). This study makes use of the latest version available, v3.0. It is a 35-year data set spanning from 1980 to 2014 and is derived from a variety of sources, namely vegetation optical depth (VOD) and snow water equivalents (SWE), satellite-derived soil moisture (SM), reanalysis air temperature and radiation and a multi-source precipitation product (Martens et
al., 2016). It is available at a spatial resolution of 0.25°. Martens et al. (2016), provide a full description of the dataset including an extensive validation using measurements from 64 eddy-covariance towers worldwide.

The up-scaled FLUXNET GPP and evapotranspiration were derived from the FLUXNET network using a model tree ensemble (FLUXNET-MTE hereafter) approach as described in Jung et al.
(2009). It is a machine learning technique that can be trained to ascertain land-atmosphere fluxes, providing a way of benchmarking LSMs at large scales (Jung et al., 2009, 2010; Beer et al., 2010; Bonan et al., 2011; Jung et al., 2011; Slevin et al., 2016 in review). The machine learning algorithm is trained using a combination of land cover data, observed meteorological data and remotely sensed vegetation properties (fraction of absorbed photosynthetic active radiation). The algorithm
uses model tree ensembles to provide estimates of carbon fluxes at FLUXNET sites with available quality-filtered flux data, after which the trained model can be implemented globally using grids of the input data (Jung et al., 2009, 2011). It is limited to a 0.5° spatial resolution and a monthly temporal resolution over a 20-year period (1982-2011). It can be found in the Max Planck Institute for Biogeochemistry Data Portal (https://www.bgc-jena.mpg.de/geodb/projects/Home.php).

## 2.3    Experimental setup

The LDAS used in this study is designed as follow; $x$ is the 8-dimensional control vector including soil layers 2 to 8 (representing a depth from 1 cm of 100cm) and LAI propagated by ISBA LSM. $y_o$ is the 2-dimensional observation vector (SSM, LAI) and the model counterparts of the observations are the second layer of soil of ISBA LSM ( $w_2$ between 1 and 4 cm) and LAI (for
SSM and LAI). A comparison between ESA CCI SM and the two top ISBA soil layers suggests that the second layer of soil better represents the satellite-derived product (not shown). Also the first





layer of soil (1 cm depth) is discarded from the control vector as over a 24-hour window it is more reactive to the atmospheric forcing than to a small initial perturbation (Draper et al., 2011, Barbu et al, 2014). This leads to the following expression of the Jacobians matrices:


$$
H = \begin{vmatrix} \dfrac{\partial SSM^t}{\partial LAI^0} & \dfrac{\partial SSM^t}{\partial w_2^0} & \cdots & \dfrac{\partial SSM^t}{\partial w_8^0} \\[2ex] \dfrac{\partial LAI^t}{\partial LAI^0} & \dfrac{\partial LAI^t}{\partial w_2^0} & \cdots & \dfrac{\partial LAI^t}{\partial w_8^o} \end{vmatrix}
$$

(9)

Several studies (e.g. Draper et al. 2009; Rüdiger et al., 2010) have demonstrated that small perturbations ($10^{-3}$ or less) lead to a good approximation of this linear behaviour, provided that computational round-off error is not significant. Following Draper et al. (2011), Mahfouf et al. (2009), the soil moisture errors are assumed to be proportional to the dynamic range (the difference between the volumetric field capacity ( $w_{fc}$ ) and the wilting point ( $w_{wilt}$ ), which is determined


by the soil texture (Noilhan and Mahfouf [1996])); in this study the Jacobian perturbations were assigned values of $1.10^{-4} \times (w_{fc} - w_{wilt})$ . Following Rüdiger et al. (2010), the LAI perturbation was set to a fraction (0.001) of the LAI itself. In this configuration, for every 24-hour analysis cycle, the LSM is run several times; first to get the model trajectory (forecast), then perturbing the


initial conditions of the various control variables, allowing computation of the various terms of the Jacobians (Eq.4).

For soil moisture in the second layer of soil, i.e. the model equivalent of the SSM observations, a mean volumetric standard deviation error of 0.04 $m^3 m^{-3}$ is prescribed. A smaller mean volumetric standard deviation error of 0.02 $m^3 m^{-3}$ is prescribed to the deeper layers, as suggested by several


authors for RZSM (Mahfouf et al., 2009; Draper et al., 2011; Barbu et al., 2011, 2014). The observational SSM error is set to 0.05 $m^3 m^{-3}$ as in Barbu et al., 2014. This value is consistent with errors estimated from a range of remotely sensed soil moisture sources (e.g. de Jeu et al., 2008; Draper et al., 2011; Gruber et al., 2016). Soil moisture observational and background errors are also scaled by the model soil moisture range, assuming that there is linear relationship between the soil


moisture errors and the dynamic range. The error standard deviations in the GEOV1 LAI and the modelled LAI (for modelled LAI values higher than 2 $m^2 m^{-2}$) are both assumed to be equal to 20% of the LAI values. In accordance with a study by Barbu et al. (2011), the modelled LAI values lower than 2 $m^2 m^{-2}$ are assigned a constant error of 0.4 $m^2 m^{-2}$.





SURFEX-CTRIP was spun up by cycling twenty times through the year 1990, then a 10-yr model

run is allowed before considering both an open-loop and an analysis experiment over 2000-2012. Prior to these runs, an analysis experiment without assimilating any observations has also been run over 2000-2012 to study the model sensitivity to the observations through the Jacobians. Studies of the Jacobian values have to be performed before assimilating observations because the validity of the linear assumptions in deriving the Jacobians is related to the sensitivity of the assimilation

system. Table 1 summarizes the SURFEX-CTRIP set-up used in this study.

*Table 1: Summary of the experimental setup used in this study.*

| Model | Domaine | Atm. Forcing | Data Assimilation Method | Assimilated Obs. | Observation Operator | Control Variables | Additional Option |
|---|---|---|---|---|---|---|---|
| ISBA model, options DIF and NIT | Europe and the Mediterranean basin (0.5°) | Earth2Observe/WFDEI | SEKF | SSM (http://www.esa-soilmoisture-cci.org) LAI (http://land.copernicus.eu/global/) | Second layer of soil (1-4cm), LAI | Layers of soil 2 to 8 (1-100cm), LAI | Coupling with CTRIP (0.5°) |

## 3 Results

### 3.1 Model and observations consistency

Observations consistency over time is crucial when assimilating long-term datasets. Several authors assessed the consistency of the ESA CCI soil moisture product with respect to re-analysis products

(e.g., Loew et. al., 2013; Albergel et. al., 2013a; 2013b) and in-situ measurements (Dorigo et. al., 2015, 2017). Lambin et al. (1999) found that the GEOV1 LAI data set is also consistent over time and can be used e.g. for detection of change and for providing information on shifting trends or trajectories in land use and cover change. To verify the results from literature for the spatial and temporal domain considered in this study a consistency evaluation both for SSM and LAI has been

performed. As observed SSM climatology is matched to the model climatology (see section 222), consistency between observations and the model over time (2000-2012) is expressed as correlations on both absolute and anomaly time-series. The latter is computed using monthly sliding windows as described in Albergel et al. (2009). Only significant correlations values (at p-value<0.005) are retained. For LAI consistency is expressed both as correlations and Root Mean Square Differences.

Median soil moisture correlation (anomaly correlation), of ESA CCI SSM with SURFEX-CTRIP



second layer of soil, $w_2$ between 1 and 4 cm, is 0.65 (0.47) over 2000-2012. Year-to-year correlation (anomaly correlation), which can potentially be impacted by the prevailing conditions in the given years, ranges from 0.62 (0.45) to 0.71 (0.48). Although many different sensors are used over time and space to retrieve ESA CCI SSM, the product can be considered stable. Over the same

period, correlation (RMSD) between GEOV1 LAI and SURFEX-CTRIP is 0.75 (0.85 $m^2m^{-2}$), correlations range from 0.72 in 2000 to 0.77 in 2012. RMSD values are relatively stable too with a minimum value of 0.76 $m^2m^{-2}$ in 2002 and a maximum of 0.91 $m^2m^{-2}$ in 2007. RMSD exhibits however a strong seasonal dependency as illustrated by Figure 2 (blue line) with values close to 1 $m^2m^{-2}$ from June to October. During these months correlation is better with values between 0.75 and

0.85. Too large RMSD values observed in winter time are not desirable since the vegetation is supposed to be dormant.

Overall both ESA CCI SSM and GEOV1 LAI were found stable over time with respect to SURFEX-CTRIP, as illustrated in Figure 3 for 2000, 2006 and 2012. Figure 3 top row illustrates correlations between ESA CCI SSM and SURFEX-CTRIP ( $w_2$ ). While in 2000 not all of Europe

is covered, it is the case from 2003 onwards. Low correlations values are found in desert areas (over the Sahara), high elevation (e.g. over the Alps) and at high latitudes whereas good values are obtained over e.g., the Iberian Peninsula, France and Turkey. Figure 3 middle and bottom rows present the correlations and RMSD values respectively for GEOV1 LAI with SURFEX-CTRIP, only for vegetated  grid points (>90%). Generally, LAI at high elevation is not represented well

(low correlations and high RMSD) as well as in the northeastern part of the domain, which is mainly covered by broad-leaves trees. Conversely, the southern part of the domain presents high level of correlations and low RMSD values.

### 3.2    Model sensitivity to observations

The Jacobians,   $H$    (Eq.4) are dependent on the model physics. Their examination provides useful

insight in explaining the data assimilation system performances (Barbu et al., 2011, Fairbairn et al., 2017). Median values over 2000-2012 are presented in Table 2.





*Table 2 : Median Jacobians values for the eight control variables considered in this study over the whole spatial domain for 2000-2012.*

| 2000-2012 | $\dfrac{\partial SSM}{\partial LAI}$ | $\dfrac{\partial SSM}{\partial w_2}$ 1-4 cm | $\dfrac{\partial SSM}{\partial w_3}$ 4-10 cm | $\dfrac{\partial SSM}{\partial w_4}$ 10-20 cm | $\dfrac{\partial SSM}{\partial w_5}$ 20-40 cm | $\dfrac{\partial SSM}{\partial w_6}$ 40-60 cm | $\dfrac{\partial SSM}{\partial w_7}$ 60-80 cm | $\dfrac{\partial SSM}{\partial w_8}$ 80-100 cm |
|---|---|---|---|---|---|---|---|---|
| Median | -0.0010 | **0.1719** | **0.1543** | 0.0694 | 0.0275 | 0.0043 | 0.0006 | 0.0001 |
| | $\dfrac{\partial LAI}{\partial LAI}$ | $\dfrac{\partial LAI}{\partial w_2}$ 1-4 cm | $\dfrac{\partial LAI}{\partial w_3}$ 4-10 cm | $\dfrac{\partial LAI}{\partial w_4}$ 10-20 cm | $\dfrac{\partial LAI}{\partial w_5}$ 20-40 cm | $\dfrac{\partial LAI}{\partial w_6}$ 40-60 cm | $\dfrac{\partial LAI}{\partial w_7}$ 60-80 cm | $\dfrac{\partial LAI}{\partial w_8}$ 80-100 cm |
| Median | **0.2220** | 0.0006 | 0.0015 | **0.0032** | **0.0068** | **0.0038** | 0.0011 | 0.0006 |

The model equivalent of SSM is the second layer of soil ( $w_2$ between 1 and 4 cm depth). It is
then expected that the sensitivity of SSM to changes in soil moisture of that layer is higher
compared to those of the other layers of soil. Sensitivity of LAI to changes in soil moisture (Table 2,
bottom rows) is generally weaker than that of SSM (Table 2 top rows) suggesting that although
control variables related to soil moisture will be impacted by the assimilation of LAI, they would be
even more impacted by the assimilation of SSM. The model sensitivity to SSM decreases with
depth as presented in Table 2 revealing that the assimilation of SSM will be more effective in
modifying soil moisture from the first layers. Over Europe, median values of $H$ with respect to
SSM observations (Table 2 top rows) range from 0.1719 to 0.0001 for layers $w_2$ to $w_8$ ,

respectively and is –0.0001 for LAI. The negative value of $\dfrac{\partial SSM}{\partial LAI}$ also indicates that a positive

increments of LAI will generally lead to a decrease of SSM ( $w_2$ ). The depth impact is also
illustrated in Figure 4 which represents histograms of $H$ over Europe for three control variables
( $w_2$ in red, $w_4$ in cyan and $w_8$ in blue) with respect to a change in SSM for six months
(January, March, June, August, October, December) over 2000-2012 (Figure 4, a to f). Additionally
Figure 4 depicts a seasonal dependency. For instance, the histogram representing $H$ of control
variable $w_2$ (Figure 4,a) presents mainly three types, (1) values close or equal to 0 (type_A), (2)
values between 0.2 and 0.8 (type_B) and (3) close to 1 (type_C). The values of type_C correspond
to the situation in which the model dynamic is close to the identity i.e. the perturbation of the initial
state is almost unchanged by the end of the assimilation window (24h). For values of type_B, the





model dynamic is strongly dissipative and therefore the final offset is only a fraction of the initial perturbation. Distributions of types A, B, C vary in time; while for January they are 75%, 14% and

11%, for June they are 36%, 44% and 20% and for October 48%, 30% and 22%, respectively. It suggests a higher sensitivity of the first layers of soil to a change in SSM, particularly during late summer and autumn than during winter months. While a similar behaviour is observed up to the fourth layer of soil, the deepest layers of soil (e.g. $w_8$, blue line) do not show any seasonal dependency, and very small sensitivity with mainly Jacobians values of type A.

The same typology can apply to $H$ values $\frac{\partial LAI}{\partial LAI}$ (Figure 4, g, h, i), with an even stronger seasonal dependency. For all Januaries, distributions are 81%, 18% and 1%, while they are 22%, 77% and 1% for Junes and 27%, 45% and 28% for Octobers for types A, B and C, respectively. Assimilation of LAI will be more effective in modifying LAI from late spring to autumn. Finally, the assimilation of LAI will be more effective in modifying soil moisture from layers 4 to 6 (Table

2) where most of the roots are present for the different vegetation types from ISBA (between 20 cm and 60 cm, see Table 1 of Decharme et al., 2013).

### 3.3   Analysis impact

Control variables are directly impacted by the assimilation of LAI and SSM, Figure 5 illustrates averaged analysis increments for the period 2000-2012 for LAI and soil moisture in $w_2$ (between

1 cm and 4 cm), $w_4$ (between 10 cm and 20 cm) and $w_6$ (between 40 cm and 60 cm) for all months of February, May, August and October. Red (blue) colours indicate that the analysis removes (adds) LAI and soil moisture. At the beginning of the year vegetation is not very active, but on the very western part of the domain the analysis tends to add LAI over the United Kingdom, northwestern parts of France and it reduces LAI over the Iberian Peninsula. At the beginning of the

year soil moisture is only slightly affected by the analysis. Later in spring and summer the analysis is more efficient: it removes LAI over a large part of Europe reducing the bias observed between open-loop and observations. It mainly adds water in $w_2$ and remove water from layers $w_4$ to $w_6$. The seasonally marked impact of the analysis is consistent with the above description of the Jacobians behaviour. Analysis increments are also presented in Figure 6 for the entire period 2000-

2012. Generally, the analysis tends to remove LAI, add water in $w_2$ but dries layers where the roots are mainly located (from $w_4$ to $w_6$). Its effect is however less pronounced at greater



depths.

Figure 7 shows the averaged analysis impact on LAI for all months of January, April, July and October over 2000-2012 expressed in RMSD in the following way: GEOV1 LAI vs. open-loop and

GEOV1 LAI vs. analysis. Only points where observed LAI is available (and assimilated) are retained. As this impact assessment is conducted against the observations that were assimilated, improvements from the analysis are expected and shows that the LDAS is working well. From Figure 7, this is mostly the case (e.g. in October). As indicated in section 3.2, the analysis is most efficient during late summer and autumn. The geographical patterns highlighted in section 3.1 are

also observed with a clear improvement, e.g. in the northeastern part of the domain. Analysis improvement with respect to the observations is also visible in Figure 7.

Figure 8 illustrates histograms of innovations (in red) and residuals (in green) of LAI for all months of February, April, June, August, October and December over 2000-2012. As expected, the distribution of residuals is more centred on 0 than the distribution of the innovations. A seasonal

pattern can be observed:  during the growing phase (and up to June) both innovations and residuals present a right tail indicating that the model (and the analysis to a lesser extent) tends to underestimate LAI. In this period, similarities between innovations and residuals suggest that the analysis is not very efficient. At the end of summer and in autumn distributions present a left tail distribution; LAI is overestimated but this time the analysis is more efficient. Distributions of SSM

residuals are even more centred on zero than those of innovations with no seasonal dependency and smaller differences (not shown). The common CDF-matching technique applied to SSM to remove systematic errors is responsible for this smaller impact as the LDAS can only correct SSM short term variability. Contrary to SSM, the LAI mismatch between the open-loop and the GEOV1 estimates concerns both magnitude and timing (see e.g. Figure 6 in Barbu et al. (2014)).

Figure 9 presents averaged differences over 2000-2012 between the open-loop and the analysis for other land surface variables that are indirectly impacted by the assimilation, namely: daily cumulated soil drainage flux, supersaturation runoff, evapotranspiration and daily mean river discharge. Although the analysis impact is relatively weak on those variables (e.g. ~1% on the river discharge represented over the Danube) geographical patterns are observed. Areas where positive

analysis increments were found for LAI (Figure 5) are marked by a decrease in drainage and runoff (in red on Figure 9) while evapotranspiration increases (in blue Figure 9). Changes in these,



indirectly impacted land surface variables are in agreement with the analysis increments maps (Figure 6).

### 3.4 Evaluation of analysis impact

First, the evaluation of the analysis impact over France is effectuated using straw cereal grain yield (GY) values from the Agreste French agricultural statistics portal. Only the '*département*' administrative units corresponding to a high proportion of straw cereals are considered. Yearly maximal above ground biomass ($B_{ag}$) values from the open-loop and analysis are compared to GY over 2000-2010. Yearly-scaled anomalies from the mean and the standard deviation for

observations, open-loop and analysis are used for 45 sites over France as in Dewaele et al. (2017). Figure 10a and b present correlations and RMSD values, respectively and Figure 10c time-series for one site illustrating the inter-annual variability. After assimilation of SSM and LAI, correlation (RMSD) between $B_{ag}$ and GY is clearly improved for 43 (41) sites out of 45 showing the added value of the analysis. Figure 10c presents $B_{ag}$ from the open-loop (black dashed line) and analysis

(black solid line) as well as observed GY (red solid line) scaled anomaly times-series for one site in Allier, France (46.0858ºN-3.21641ºE). Correlations (RMSD) for open-loop and analysis experiments are 0.45 (0.99) and 0.78 (0.63), respectively.

Over 2000-2010, 48 of 83 gauge station present $Eff.$ values greater than 0, 22 greater than 0.5. As suggested in the previous section, the analysis impact on river discharge is rather small. If the

analysis generally leads to an improvement in river discharge representation, only 8 stations present an $Eff.$ increase superior to 0.05 (3 present a decrease superior to 0.05). $Eff.$, R and RMSD histograms of differences are presented in Figure 11 along with a hydro-graph for the Loire River in France (47.25ºN, 1.52ºW). Although the assimilation impact is relatively small, evaluation results suggest that they are neutral to positive. Analysis impact on other CTRIP variables (e.g., floodplain

fraction and storage, groundwater height) is rather neutral.

Evapotranspiration from both the open-loop and the analysis are compared to monthly values of GLEAM over 2000-2012 for vegetated grid points (>90%). As for the river discharge, the assimilation impact on evapotranspiration is small. However the comparison with the GLEAM satellite-derived estimates is rather positive, as illustrated in Figure 12 representing

evapotranspiration from the open-loop (Fig.12a) , GLEAM estimates (Fig.12b), the analysis (Fig.12c) and their differences (Fig.12d) and in Figure 13 showing differences in RMSD (Fig.13a)





and correlations (Fig.13b) between the open-loop (analysis) and GLEAM estimates, respectively. Open-loop simulation of evapotranspiration tends to over-estimate the GLEAM product over most of Europe, particularly over France and the Iberian Peninsula, North Africa. Analysis is able to reduce this bias (Figure 12d). Most of the differences in RMSD and correlations are negative and positive: 76% and 80% showing the added value of the analysis.

As for evapotranspiration, GPP from both the open-loop and the analysis are compared to monthly GPP estimates from FLUXNET-MET dataset. Figure 12 illustrates averaged carbon uptake by GPP over land for 2000-2011 from the open-loop (i.e model) (Fig.12e), FLUXNET-MET (Fig.12f) and the analysis (Fig.12g) as well as differences between the analysis and the model (Fig.12h). Also, Figures 13 c) and d) show RMSD and correlation differences between the open-loop or the the analysis and FLUXNET-MET dataset (analysis minus open-loop). Finally Figure 13 presents seasonal scores over the same period (RMSD and Correlation values). Compared to the FLUXNET-MTE estimates, the open-loop tends to underestimate GPP over the Scandinavian countries, the northwestern part of France, UK and Ireland, north of the Caspian Sea while an overestimation is visible over most of the Iberian peninsula, Eastern Europe as well as the north-eastern part of the domain (Figure 13, a, b). From Figures 13 d) and e) and Figure 13 one may notice that after assimilation of SSM and LAI there is a clear improvement in the GPP representation for RMSD and correlation with a systematic seasonal decrease and increase of the scores. Over the whole domain, 79% and 90% of the grid points present better RMSD and correlation values after assimilation with respect to the FLUXNET-MTE estimates of GPP.

Evapotranspiration from the open-loop and analysis has also been evaluated using FLUXNET-MET estimates of evapotranspiration. Results are illustrated by Figure 12i to l and Figure 13e and f. They are similar of those obtained using GLEAM estimates: over the whole domain most of the differences in RMSD and correlations are negative and positive: 70% and 79%.

## 4 Discussion

### 4.1 Can different data assimilation techniques improve the analysis?

This study introducing the LDAS-Monde is based on a Simplified version of an Extended Kalman filter. Although a version of an Ensemble Kalman Filter is available (EnKF, Evensen, 1994), to date SEKF is the most mature technique developed for land surface data assimilation within SURFEX.





Many studies using SURFEX exposed the strengths and weaknesses of this technique (Mahfouf et. al., 2009, Albergel et. al., 2010., Draper et. al., 2011, Barbu et al., 2011, 2014, Duerinckx et. al., 2015, Fairbairn et. al., 2015, 2017). The SEKF relies on accurate linear assumptions in deriving the Jacobians. Draper et al. (2009), Duerinckx et. al. (2015) and Fairbairn et al. (2015) pointed out that

excessive Jacobians may occur under specific conditions (e.g. close to threshold values like the wilting point and field capacity for soil moisture) possibly leading to instabilities in the analysis. They were however obtained using the force-restore version with three layers of soil. In such configuration they used only one control variable related to soil moisture; the second layer of soil that is a thick layer representing all the root-zone ( $w_{2-RZ}$ ) while the observation operator is the

very thin top layer (~ 1cm). Thus $\dfrac{\partial SSM}{\partial w_{2-RZ}}$ Jacobians, representing the impact of perturbing $w_2$

(i.e. the whole root-zone) on SSM (~ 1cm) can be quite different compared to those obtained using the soil diffusion scheme and presented in this study (e.g., where $w_2$ and SSM representing the same depth; 1-4 cm). For instance, they exhibit a rather large proportion of negative values as illustrated by Figure 10 of Fairbairn et al. (2017) and discussed in Parrens et al. (2014). Very few

negative Jacobian values are obtained with the diffusion soil scheme (as in Figure 4) over Europe for 2000-2012. The SEKF is also limited in correcting errors from the atmospheric forcing uncertainty making the system relying too much on the chosen forcing. Alternatively an EnKF, which relies on the ensemble spread to capture background errors, can be modified to stochastically capture both model and precipitation errors (Maggioni et al., 2012; Carrera et al., 2015). The use of

an EnKF within LDAS-Monde is currently under investigation at Meteo-France. Alternatively, particle filters could provide a means to capture non-Gaussian errors (e.g., Moradkhani et al., 2012).

The performance of an analysis scheme depends on appropriate statistics for background and observation errors. Wrongly specified error parameterisation may negatively affect the analysis. The main objective of this study was to present the newly developed LDAS-Monde while the statistics

for background and observation errors were obtained from the literature. Soil moisture observations and background errors were scaled using the model dynamic range. It accounts for texture-based spatial variability in the error and assumes that the soil moisture errors and the dynamic range have a linear relationship. Time correlations in the errors have also been neglected in this study, which are likely to occur in reality. It is also possible to employ an a-posteriori diagnostic to estimate





observation errors, such as the statistics of the innovations (observations-minus-background)
(Andersson, 2003; Mahfouf et al., 2007). This approach has been successfully applied by Barbu et
al. (2011) on a point scale experiment to obtain seasonal error variability, the approach does not
provide objective estimates of the observational errors but assesses the sub-optimality of the
analysis. Future work will investigate having spatially and temporally variant observations errors,

based on elaborated methods already applied to the ESA CCI SSM dataset like triple-collocation
(Dorigo et al., 2015) or error decomposition (Su et al., 2016).

Having LAI estimates every 10 days while using 24h assimilation window may also trigger analysis
discrepancies, as between two LAI assimilations the system relies only on SSM assimilation. When
a large analysis update occurs on LAI (from the assimilation of LAI), it then tends to go back

towards the model states in the successive days before being constrained again by the next

observations. For instance, in winter most of the $\frac{\partial LAI}{\partial LAI}$ Jacobians are equal (or close) to zero and

therefore the analysis update caused the LAI to return almost instantaneously to the incorrect LAI
minimum value. This issue could be addressed using longer assimilation windows, from 10 days up
to one month (e.g. as in Jarlan et al., 2008) where different data assimilation techniques could be

used (e.g. variational methods to obtain a best fit between several observations). An alternative
could be to keep a 1-day assimilation window and use smoothing techniques (e.g. Munier et al.,
2014) to keep the benefit of the analysis update by propagating the error covariance forward up to
the next available observation.

    4.2    Can a better use of the observations improve the analysis?

610        4.2.1    Towards a better use of GEOV1 Leaf Area Index

SURFEX_v8.0 does not use any crop-specific parameterisation, which would be required to
simulate the crop grain yield formation. In addition, the simulations of photosynthesis and
vegetation growth do not take into account certain factors impacting the long-term agricultural
production (e.g., changes in agricultural practices, diseases, pests, crop migration, the grain

formation and crop cultivars). However, previous studies (Calvet et al., 2012, Canal et al., 2014)
showed that agricultural statistics like grain yields can be used to benchmark SURFEX
development in representing the above ground biomass inter-annual variability. A strong positive
impact from the assimilation of SSM and LAI on the representation of above ground biomass inter-





annual variability has been highlighted in this study. The impact on river discharge representation is
only small (section 3.3). Improvements are however expected from a better representation in the
model of vegetation parameter like LAI (e.g., Szczypta et al., 2014). Although the analysis is
efficient in correcting LAI, high RMSD values remain, particularly during the senescence phase
when SURFEX-CTRIP over-estimate LAI over a large part of Europe. RMSD and correlations with
GEOV1 and SURFEX-CTRIP still expose a strong seasonal dependency after the analysis (red line
on Figure 2) which is mainly attributed to model errors. The GEOV1 estimates have been shown to
exhibit some realistic environmental features that are not, or poorly, simulated by the model
(Fairbairn et al., 2017). Therefore, it was decided not to rescale the GEOV1 estimates to the model
climatology.

Figure 2 also suggests that the minimum LAI values used as model parameters (see section 2.1.1)
should be revisited because too large differences are not desirable particularly when the vegetation
is dormant. Another caveat of this study is the use of a single LAI value for all vegetation types that
are represented in SURFEX-CTRIP. As detailed in Barbu et al. (2014), the Kalman gain is
calculated for each individual vegetation type (patch). The analysis increment is added to the
background for each patch, producing a patch-dependent analysis update. The patch-dependence is
introduced in the Kalman gain via the Jacobian elements. The possibility of having LAI estimates
for each type of vegetation is under investigation and has the capacity of overcoming the two
above-mentioned weaknesses. Recently, the GEOV1 LAI data has been disaggregated following a
Kalman filtering technique developed by Carrer et al. (2014). This enables the LAI signal for each
patch to be separated within the pixel, which provides a dynamic patch-dependent estimate of the
assimilated LAI within the pixel (Munier et al., 2017, in prep.). From the individual estimates over
1999-2015, minimum LAI values have also been used to update the model parameterisation.
Preliminary results from assimilating disaggregated LAI time series and using new LAI minimum
values (not shown) suggest an added value on vegetation variables like above-ground biomass and
on the representation of river discharge. Better performances from the assimilation of disaggregated
LAI are also expected on the representation of evapotranspiration.

### 4.2.2   Towards a better use of microwave satellite observations for soil moisture

ESA CCI SM is based on multiple microwave sources from space, namely passive radiometer
brightness temperature ($T_b$) observations and active radar backscatter ($\sigma_o$) observations. As they





are both indirectly related to soil moisture, retrieval methods making use of e.g. radiative transfer

model (for $T_b$, Kerr et al., 2012) or change-detection approaches (for $\sigma_o$, Wagner et al., 1999) are

usually required to transform Tb and $\sigma_o$ into soil moisture values that can be assimilated in

LSMs. Despite the proven record of assimilating soil moisture retrieval from point scale to regional

and continental scale (e.g. Albergel et al., 2010 ; Draper et al., 2012; Matgen et al., 2012; De

Rosnay et al., 2013; Barbu et al., 2014; Wanders et al., 2014; Ridler et al., 2014), there is an

increasing tendency towards the direct assimilation of Tb and $\sigma_o$ observations (De Lannoy et al.,

2013; Han et al., 2014; Lievens et al., 2015, Lievens et al., 2017). Retrieval methods usually make

use of land surface parameters and auxiliary information, like vegetation, texture and temperature,

possibly proving inconsistencies with specific model simulations   (which also include these

parameters but potentially from different sources). Also, if retrievals and model simulations rely on

similar types of auxiliary information, their errors may be cross-correlated, potentially degrading

the system performance (De Lannoy and Reichle, 2016). The direct assimilation of Tb and $\sigma_o$

observations requires that the LSM is coupled to a radiative transfer model that serves as a forward

operator for predicting $\sigma_o$ and/or $T_b$. It has the advantage of allowing for consistent parameters

and auxiliary inputs between the model simulations and the radiative transfer model, avoiding

cross-correlated errors. The development of a forward operator for $\sigma_o$ from active microwave

instruments is under-way at Meteo-France, it will allow accounting for vegetation effects in the

signal and using the vegetation information content of $\sigma_o$ .

## 5  Conclusions

This study provides an assessment of the LDAS-Monde implementation to increase monitoring

accuracy for land surface variables over the Europe-Mediterranean area. Satellite-derived surface

soil moisture and leaf area index are assimilated over 2000-2012 in the $CO_2$-responsive and

multilayer diffusion scheme version of the ISBA land surface model coupled with the CTRIP

hydrological system. Joint assimilation of leaf area index and surface soil moisture has been shown

to efficiently improve the representation of above-ground biomass, gross primary production and

evapotranspiration, while having a neutral to positive impact on river discharge. To our knowledge,

LDAS-Monde is the only system able to sequentially assimilate vegetation products together with

soil moisture observations. LDAS-Monde permits an efficient monitoring of various land surface

variable and has a powerful potential in monitoring extreme events like agricultural droughts at a



global scale.

The analysis of the Extended Kalman Filter observation operator Jacobians permitted to identifying both seasonal and soil depth effects of the assimilation on ISBA. A clear added value of the assimilation has been highlighted based on agricultural statistics over France, evapotranspiration and gross primary production observations based estimates over the whole domain. More analysis impact could however be expected on variables like river discharge. Studies focusing on a better use

of the observations along with other data assimilation techniques like the Ensemble Kalman Filter are currently under-way. Recent studies discussed in the previous section suggest that the direct assimilation of microwave observations of Tb and $\sigma_o$ instead of Level 2 or 3 soil moisture products could leads to better results. The development of a forward operator for $\sigma_o$ from active microwave instruments is under-way. The long-term confrontation of model and observations at

continental scale prior to the assimilation has also highlighted parameterisation issues like the minimum leaf area index values used as threshold when the vegetation is dormant. The GEOV1 leaf area index estimates permit setting up new thresholds for the different vegetation patches used in ISBA thanks to the development of a disaggregated product resulting to new leaf area index estimates, different for each patch. The assimilation of this new product is also promising.


**Acknowledgement – The work of Simon Munier was supported by European Union Seventh Framework Programme (FP7/2007-2013) under grant agreement no. 603608, "Global Earth Observation for integrated water resource assessment" (eartH2Observe). The work of Hélène Dewaele was supported by CNES and by Météo-France. The work of Emiliano Gelati was**

**supported by the French REMEMBER project (ANR 2012 SOC&ENV 001) within the HYMEX initiative. Wouter Dorigo is co-funded by the "TU Wien Wissenschaftspreis 2015" a personal grant awarded by the Vienna University of Technology. The authors acknowledge ESA's Climate Change Initiative for Soil Moisture (Contract No. 4000104814/11/I-NB and 4000112226/14/I-NB) and the EU FP7 EartH2Observe project (grant agreement number 331**

**603608) for supporting the development and evolution of the ESA CCI SM product. The authors would like to thank the Copernicus Global Land service for providing the satellite-derived LAI products.**





**Code availability**

LDAS-Monde is a part of the ISBA land surface model and is available as open source via the surface modelling platform called SURFEX. SURFEX can be downloaded freely at http://www.cnrm-game-meteo.fr/surfex/ using a CECILL-C Licence (a French equivalent to the L-GPL licence; http://www.cecill.info/licences/Licence_CeCILL-C_V1-en.txt). It is updated at a relatively low frequency (every 3 to 6 months). If more frequent updates are needed, or if what is

required is not in Open-SURFEX (DrHOOK, FA/LFI formats, GAUSSIAN grid), you are invited to follow the procedure to get a SVN account and to access real-time modifications of the code (see the instructions at the first link). The developments presented in this study stemmed on SURFEX version 8.0 and are now part of the version 8.1 (revision number 4621).





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



**Figures**

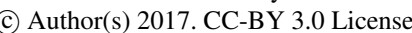

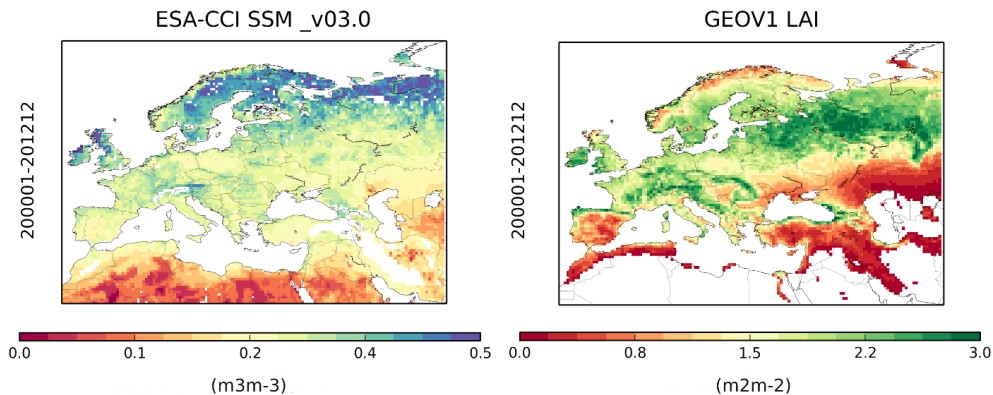


*Figure 1: Averaged (left) surface soil moisture from the Climate Change Initiative project of ESA (rigth) GEOV1 Leaf Area Index from the Copernicus Global Land Service project (for pixels covered by more than 90% of vegetation) over 2000-2012*

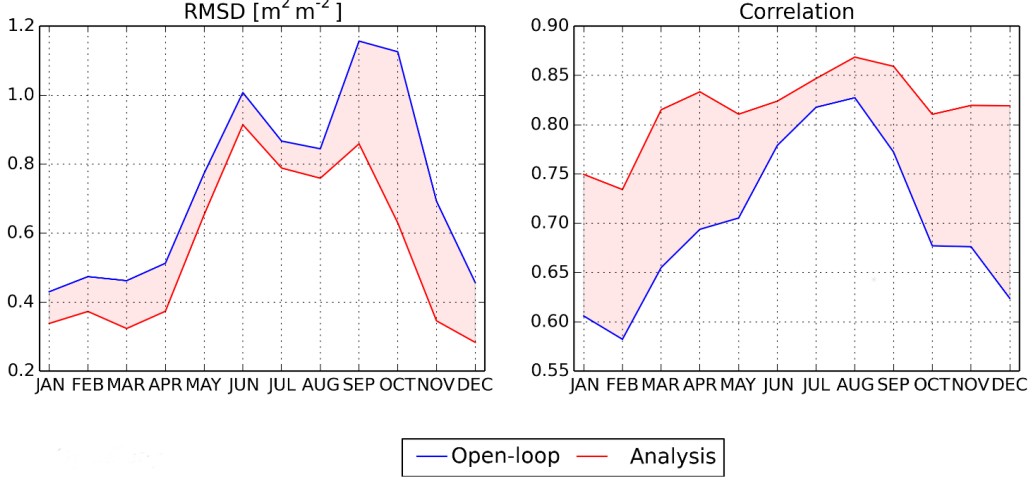

*Figure 2: Monthly RMSD and correlation values between leaf area index (LAI) from the open-loop, analysis and GEOV1 LAI estimates from the Copernicus Global Land Service project over 2000-2012*





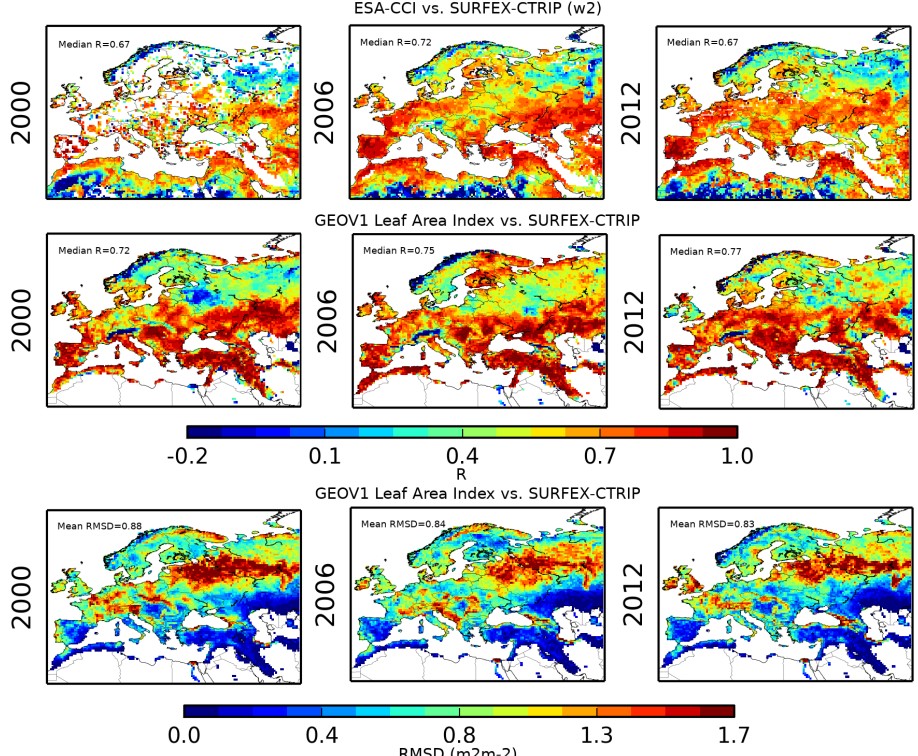

*Figure 3 : top row, correlations between satellite-derived surface soil moisture from the Climate Change Initiative project ESA and the second layer of soil (1 cm-4 cm depth) of SURFEX-CTRIP for 2000, 2006 and 2012. Middle row, same as top row for the GEOV1 leaf area index from the Copernicus Global Land Service project and SURFEX-CTRIP. Bottom row, same as middle row for RMSD. Averaged values are reported on maps.*







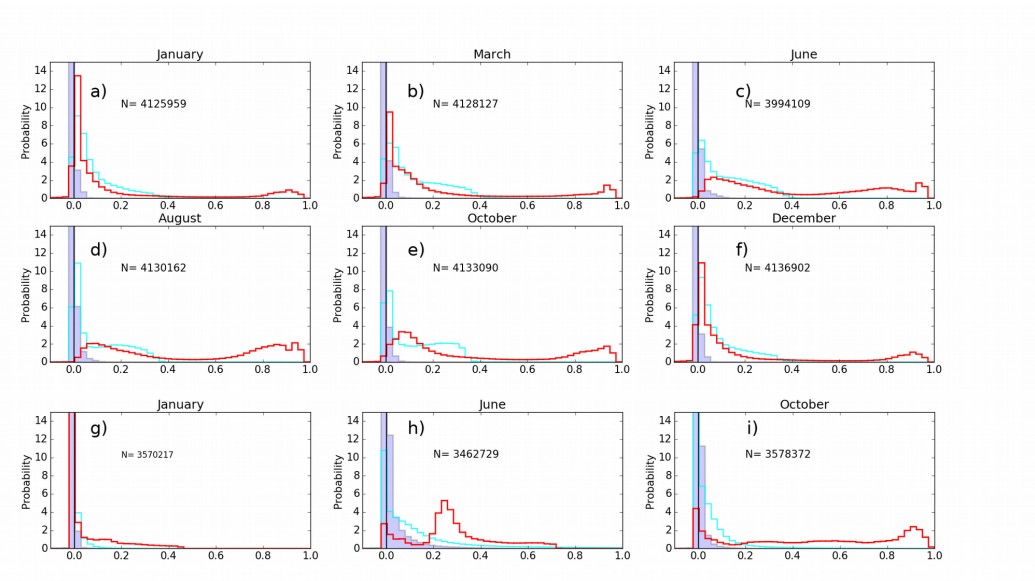

*Figure 4: Jacobian values distribution: a) to f),* $\dfrac{\partial SSM}{\partial w_2}$ *(red line),* $\dfrac{\partial SSM}{\partial w_4}$ *(cyan line) and* $\dfrac{\partial SSM}{\partial w_8}$ *(blue line) all months of January, March, June, August, October and December over 2000-2012, g) to i),* $\dfrac{\partial LAI}{\partial LAI}$ *(red line),* $\dfrac{\partial LAI}{\partial w_4}$ *(cyan line) and* $\dfrac{\partial LAI}{\partial w_8}$ *(blue line) for all months of January, June and October over 2000-2012.*



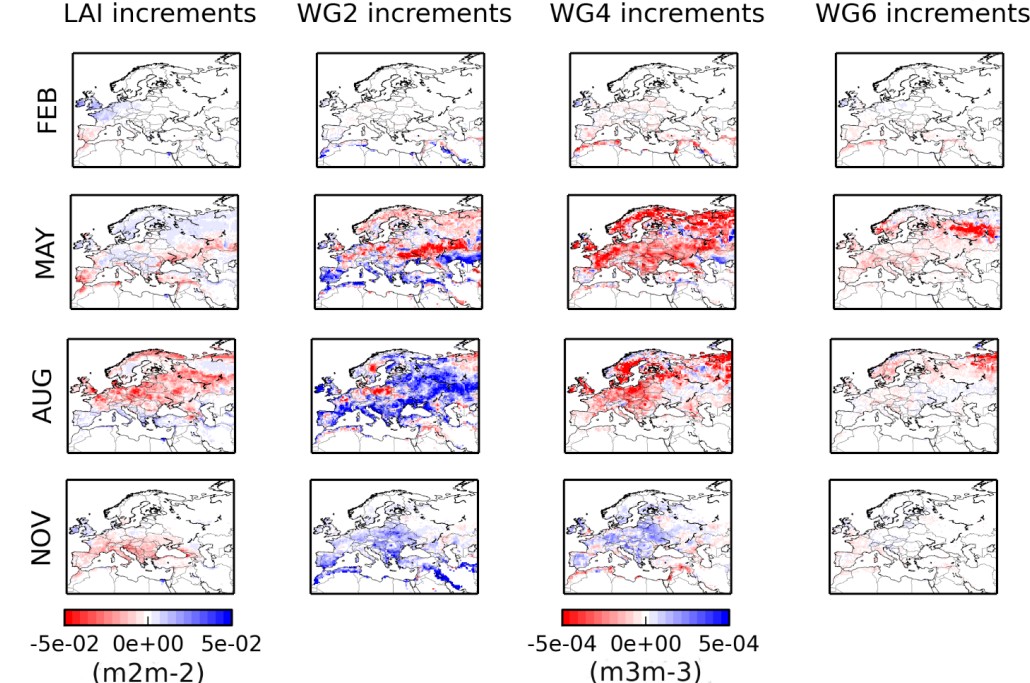


*Figure 5 : Averaged analysis increments for all months of February, May, August and November over 2000-2012 for 4 control variables; leaf area index and soil moisture in the second (1 cm- 4 cm), fourth (10 cm-20 cm) and sixth (40 cm – 60 cm) layer of soil, respectively.*





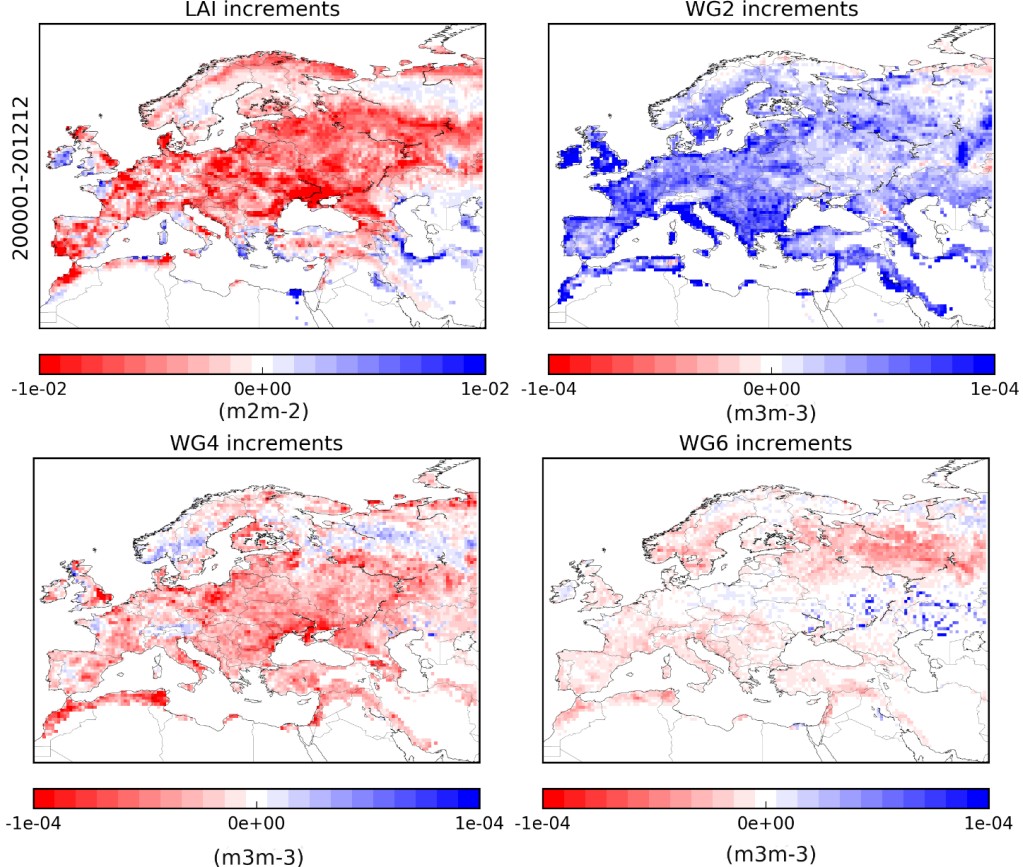

*Figure 6 : same as figure 5 for the whole 2000-2012.*






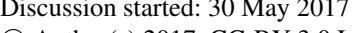


*Figure 7: RMSD maps between leaf area index from the open-loop (analysis) and that from the Copernicus Global Land Service project (GEOV1 index) for January, April, July and October over 2000-2012.*





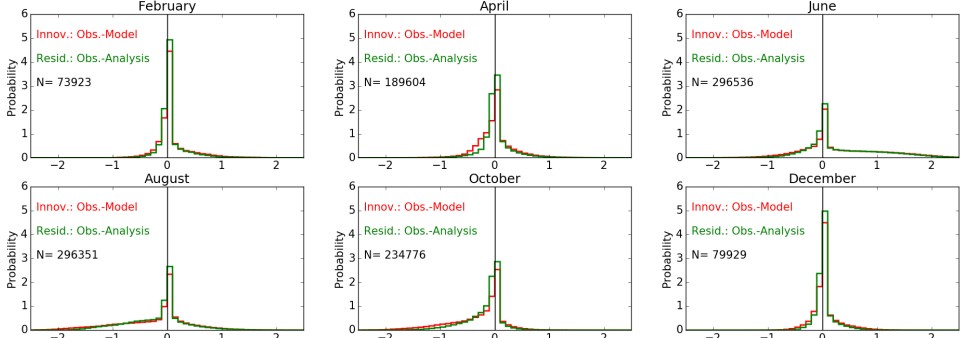

*Figure 8 : Histograms of innovation (observations-open-loop in red) and residuals (observations – analysis, in green) for Leaf Area Index for February, April, June, August, October and December over 2000-2012*



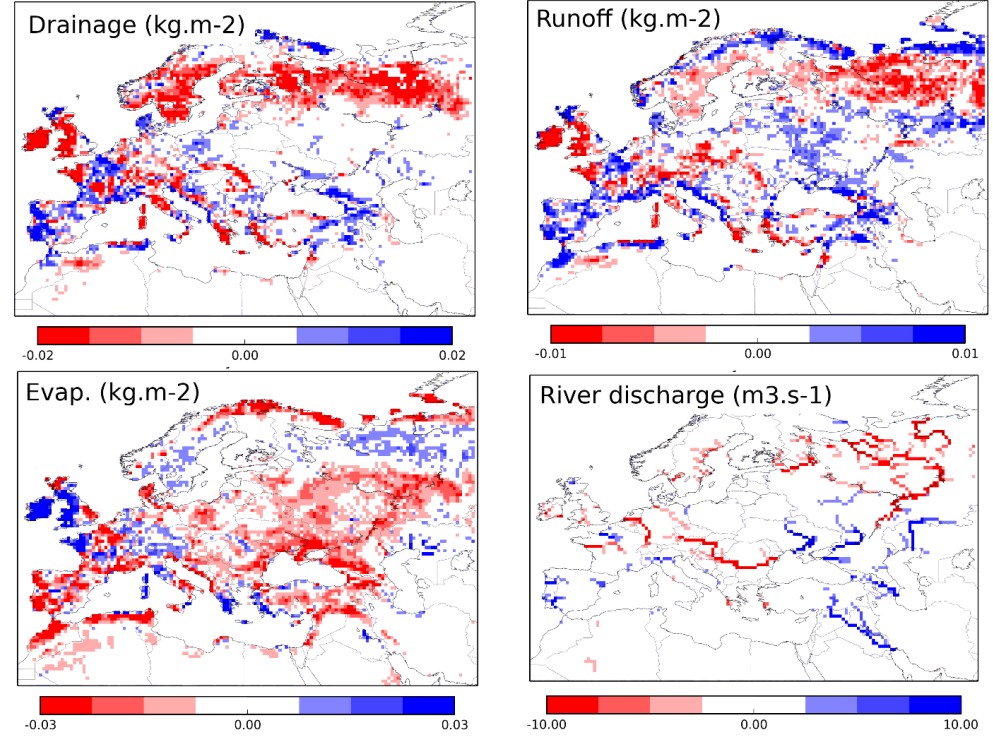

*Figure 9: Averaged analysis impact on land surface variables that are indirectly affected over the period 2000-2012.*





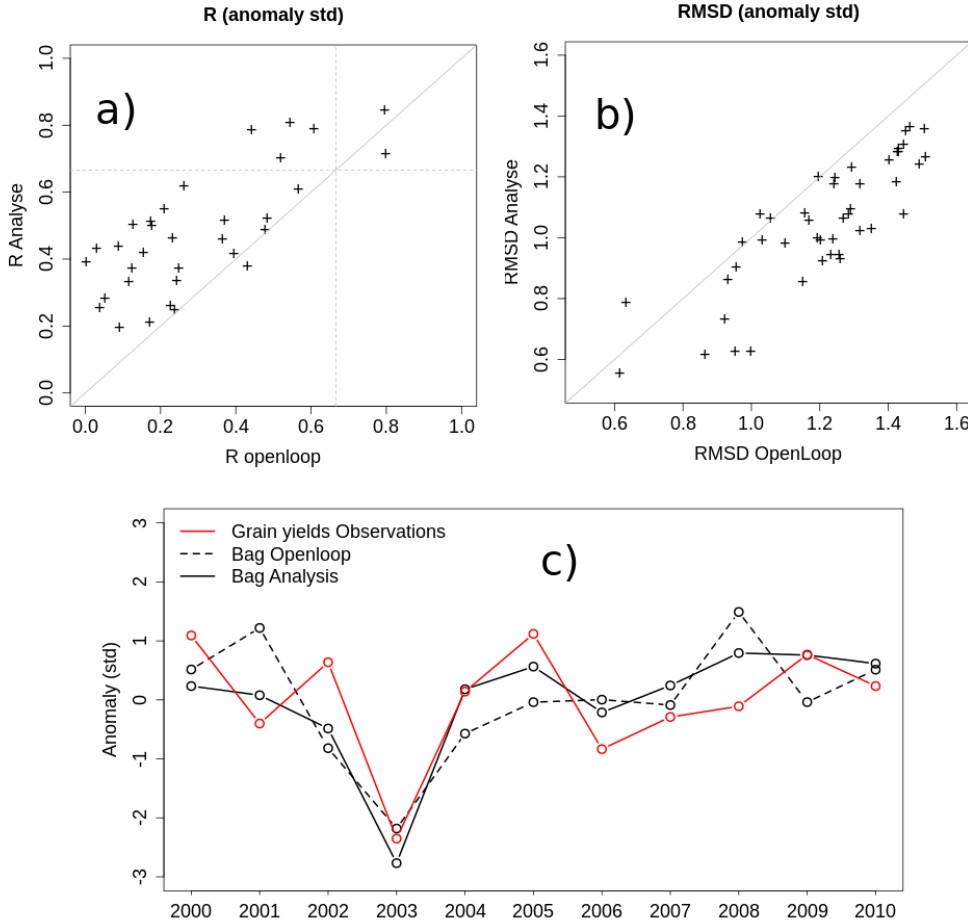

*Figure 10 : a); Correlation values for the above ground biomass from the open-loop with grain yields estimates from Agreste agricultural statistics over 45 sites in France plotted against correlations between the same quantities but above ground biomass from the analysis; b) same as a) for RMSD values; c) scaled anomalies time-series of above ground biomass from the open-loop (black dashed line) the analysis (black solid line) and grain yields observations (red solid) for one site in Allier, France (46.0858ºN-3.21641ºE).*




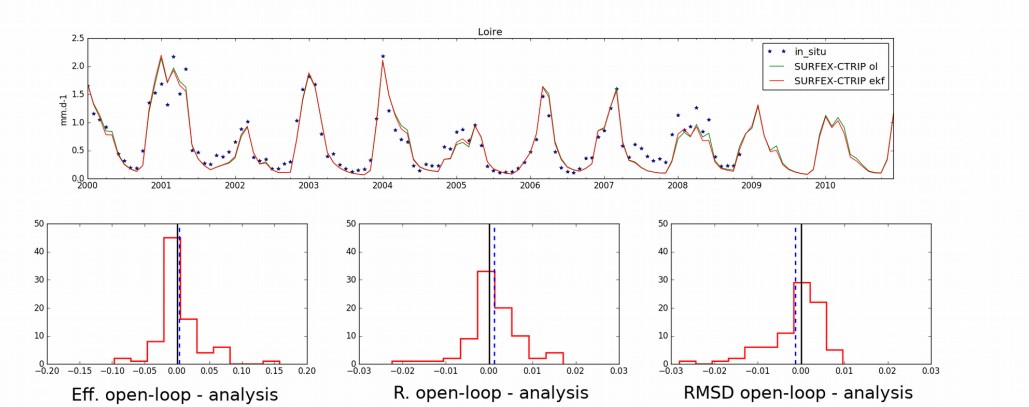

*Figure 11 : a) hydrograph for the Loire River in France (47.25ºN, 1.52ºW) representing scaled river discharge Q (using either observed or simulated drainage areas), in situ data (blues dots), open-loop (green solid line) and analysis (red solid line); b) to d) histograms of Efficiency, Correlations and RMSDs differences between Q from the open-loop and the analysis compared to the observations for the 83 stations retained (see section 2.2.3 on evaluation strategy).*





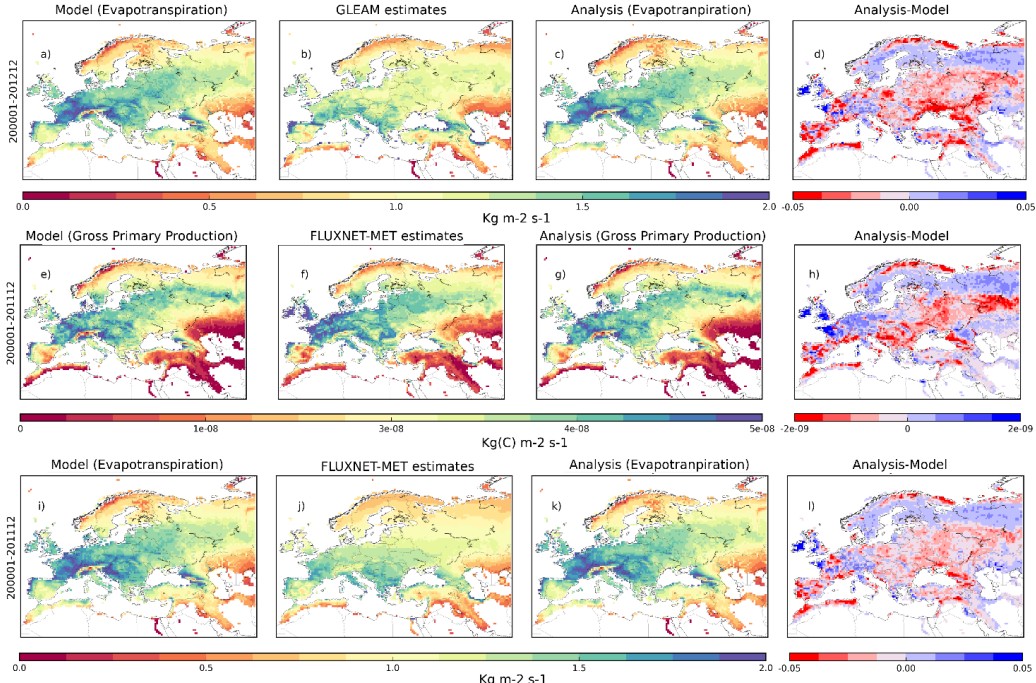


*Figure 12 : a), b), c) and d) Averaged evapotranspiration from the model (i.e open-loop), the GLEAM estimates the analysis and differences between the analysis and model over 2000-2012, respectively. e), f), g) and h) same as a), b), c) and d) for averaged carbon mass flux out of the atmosphere due to Gross Primary Production from the model, FLUXNET-MET GPP estimates the analysis and differences between the analysis and model over 2000-2011. i), j), k) and l) same as a), b), c) and d) averaged evapotranspiration from the model, FLUXNET-MET evapotranspiration estimates the analysis and differences between the analysis and model over 2000-2011.*





*Figure 13: a) and b) RMSD and Correlations differences between analysed (modelled)*




*evapotranspiration and GLEAM estimates over 2000-2012, c) and d) same as a) and b) for Carbon mass flux out of the atmosphere due to Gross Primary Production from the analysis (model), and FLUXNET-MET GPP estimates over 2000-2011, e) and f) same as a) and b) for analysed (modelled) evapotranspiration and FLUXNET-MET evapotranspiration estimates over 2000-2011.*

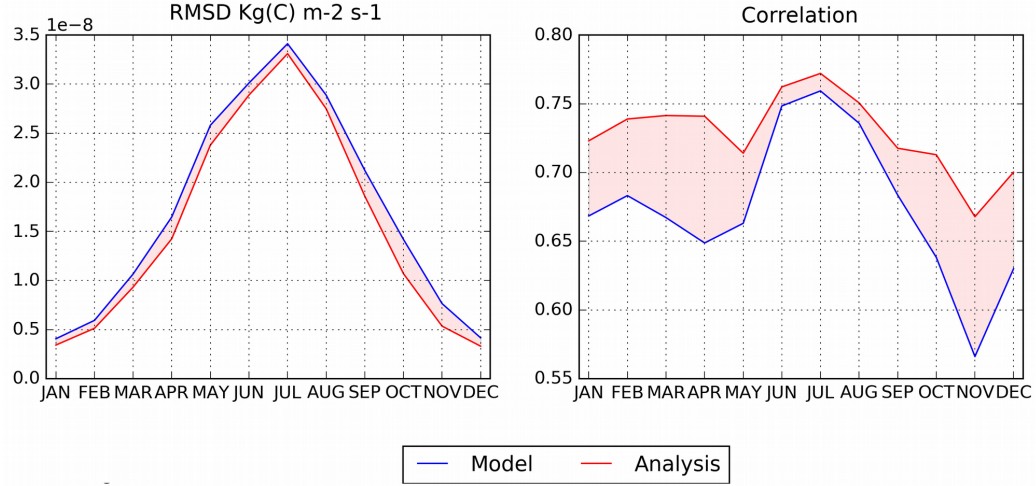

*Figure 14: As figure 2 for Carbon mass flux out of the atmosphere due to Gross Primary Production on land over 2000-2011.*