# Peer review of "Sequential assimilation of satellite-derived vegetation and soil moisture products using SURFEX v8.0: LDAS-Monde assessment over the Euro-Mediterranean area"

_Geoscientific Model Development, 2017_

## Referee Comment (RC1) · Anonymous Referee #1 · 14 Jul 2017

Review on the paper entitled "Sequential assimilation of satellite-derived vegetation and soil moisture products using SURFEX_v8.0: LDAS-Monde assessment over the Euro-Mediterranean area" by Albergel et al.

This paper presents the LDAS-Monde data assimilation system and it evaluate results over Europe. Results are well presented, showing interesting consistent impact of the data assimilation on soil moisture, LAI and hydrological variables such as discharge. It is a innovative study of high interest for the community because it presents multi-variate data assimilation in a coupled land surface and river routing model. The paper is very

clearly written and very well presented. I recommend the paper to be published after the minor comments below are accounted for.

Page 1, lines 12-13: The first sentence of the abstract should be reformulated to clarify the two objectives of the paper which are testing LDAS-Monde and improving monitoring. It does not make sense to says that LDAS-Monde "is tested ... to increase monitoring accuracy...". Testing itself only allows to evaluate and to asses monitoring accuracy.

Page 2 line 31: Replace "Assimilation impact shows that" by "results show that"

Page 2 line 33: "The assimilation impact's evaluation is succesfully carried out using...". It is not clear what succesfully means here. Is it that it worked technically or that it is a comprehensive evaluation using different data sets, or the results show good performance?. I would just replace by "A comprehensive evaluation of the assimilation impact is conducted using ..."

Page 6, line 147: replace "depth" by "deep"

Page 8 equation 5: it is not clear what ti and ti+1 are. ti must be the analysis time, but "+1" needs to be clarified: time step? analysis window length?

Page 8, 219: "WFDEI originates from the ECMWF ERA-Interim reanalysis (Dee et al., 2011) with a spatial resolution of 0.5": replace with by "interpolated to" otherwise it gives the impression that ERA-Interim is at 0.5 degrees resolution which is wrong.

Page 9, line 3: reformulate the sentence to avoid parenthesis (confusing because they do not correspond to mathematical formulation in this case): Where SSMm, SSMo, $\sigma$m and $\sigma$o correspond to the model and observation means and standard deviations, respectively.

Page 10 line 270: replace "background" by "equivalent of the observation"

Page 11 line 310: Here the time is a mean time for a given month, whereas earlier

in the paper "t" was used for instantaneous time. Replace "time t" by "month mt" or something like that.

Page 11 line 310, and page 19 lines 518, 521: replace " Eff. " by "Eff"

Page 13 line 356: "))"

Page 13 lines368-370: "Soil moisture observational and background errors are also scaled by the model soil moisture range, assuming that there is linear relationship between the soil moisture errors and the dynamic range. This was already said lines 354-356. Avoid repetition.

Page 14 table1: "Earth2Observe" by "EartH2Observe" and define NIT.

Page 14 line 377-380: This is not clear. The Jacobian values study could be done using the background forecasts of the analysis experiment. Please clarify.

Page 14 line 390: "section 222" by "section 2.2.2"

Page 14 392-395 and figure 2: It is not clear how the monthly LAI correlation are computed with one observation every ten days.

Page 16 lines 426-429: "Sensitivity of LAI to changes in soil moisture (Table 2, bottom rows) is generally weaker than that of SSM (Table 2 top rows) suggesting that although control variables related to soil moisture will be impacted by the assimilation of LAI, they would be even more impacted by the assimilation of SSM." It is not possible to directly compare jacobians values of dSSM/dW (top row) and dLAI/dw (bottom) as they don't have the same unit at all. So, the logic of this sentence does not work. The authors should elaborate the analysis of the table results to come to the conclusion that SSM assimilation should have more impact than LAI assimilation.

Page 17: lines 451-452: just say "January", "June" and "October" to be consistent with the wording used just above (eg line 444) to present this figure. The figure caption makes it clear that it is for the multi-year period.

Page 18: line 500: "Areas where positive analysis increments were found for LAI (Figure 5) are marked by a decrease in drainage and runoff (in red on Figure 9) while evapotranspiration increases (in blue Figure 9)." It is not that systematic: drainage and runoff impact is more patchy than LAI increments. I would replace "are marked by" by "tend to correspond to"

Page 19, line 516 and Figure 10 caption: "46.0858 N-3.21641E" is it necessary to give 5 digits? It is finer than the model resolution and if the purpose is the traceability of the observed GY location is there an site id number that could be used instead? It would be clearer in the paper to stick to two digits latitude-longitude information (as in line 523 for the discharge).

Figure 14: not discussed in the paper

Page 20 line 552-555 and Figure 12: I would expect figure 12 d and l to be identical as they are both indicated to show 'analysis-Model' evapotranspiration. Please clarify.

Page 21: line 569: 'the observation operator is the very thin top layer' this is not correct. Replace by 'the model equivalent was the very thin top layer'. Also replace "that is a thick layer" by "that was a thick layer" to be consistent with the first part of the sentence line 568.
* * *

---

## Referee Comment (RC2) · Anonymous Referee #2 · 26 Jul 2017

General Comments:

In the paper "Sequential assimilation of satellite-derived vegetation and soil moisture products using SURFEX_v8.0: LDAS-Monde assessment over the Euro-Mediterranean area", the authors are motivated "to increase monitoring accuracy" of land surface variables such as soil moisture (SM) and leaf area index (LAI) over the European Mediterranean region. They use a global Land Surface Assimilation system called "LDAS-Monde" to assimilate both SM and LAI in experiments to assess the effects on the model land surface variables. The model results are compared to indepen-

dent observations of river discharge, land evapo-transpiration and different agricultural statistics and measures.

I recommend that this paper undergo major revisions.

Specific Comments:

The last few sections are not clearly organised or written. The abstract and introduction are very clear about the purpose of the paper, however, the sections from 3.4 onwards lack clarity and do not lead the reader to a direct interpretation of the results as stated in the abstract. The reader would have difficulty coming to the conclusions that are put forward in the abstract. I recommend that these sections be carefully re-written.

The title includes acronyms that should be spelled out in full if they really need to be used at all. Not everyone is familiar with the acronym "LDAS" or "SURFEX". Is it necessary to put a specific version number of "SURFEX" in the title?

In many places the RMSD and correlations computed are discussed in the same sentence and this creates confusion. It would be simpler to have two or more shorter sentences that are more explicit about which measure is being used for the comparison. I think that overall the authors have chosen brevity over clarity.

Also, the sign (positive or negative) of a change in the metric used is given without explaining what the change means in terms of the variables or the models. A physical interpretation of such results would be helpful.

Please rewrite the sentence in 535-536 "Most of the differences in RMSD are negative..." RMSD is a strictly positive or zero value. Are the differences in RMSD between two different data sets being compared? Could the authors please write two sentences that explain this point more explicitly? It appears to be an important point as it is going to "show the added value of the analysis".

The figures (details are given in the technical comments below) need work as well. For example, in Figure 8: What is N? You don't really need a legend for the red and

green lines on each of the 6 month plots. Just define this in the caption. Panels need labels a, b,c and they need to be referenced as such in the caption. Please label the x axis with variable name and units. Most importantly, the y-axis is not a probability but a frequency of occurrence (the caption even says "histogram" which is correct). The integral of the probability distribution function should be equal to one by definition.

The last two paragraphs of section 3.4 are very unclear. The sentence "From Figures 13 d and e and Figure 13 one may notice...seasonal decrease and increase of the scores." does not make sense. Perhaps the authors need several sentences here that are more precise about which figures support which conclusions. Again, the RMSD and correlation should be discussed separately to make the evidence clearer. Line 555 contains another confusing sentence. "differences in RMSD and correlations are negative and positive: 70% and 79%. This just doesn't make any sense. Are there percent changes in a particular direction? If so, what are the implications for the model output or the physical system?

Section 4.1 is called "Can different data assimilation techniques improve the analysis?" I don't believe that this particular question has been answered in this section by the work presented here. I think that alternative methods are proposed and discussed but the actual results in the paper do not answer the question whether one techniques is better than another. If the section could be renamed, that would be more clear.

Technical Corrections:

Abstract: SM is defined but later in the main body SSM is used. Perhaps should just define SSM in abstract and stick to it?

Introduction: Define acronyms MODCOU and SAFRAN

Section 3.1 Should be "Consistency between the model and observations"

Need to state what is consistent with what. "Observations consistency over time..." is not clear. Do you mean that one set of observations are consistent with another

observation set? or with the model output?

lines 61-63: should read "perform best for plant productivity...to used soil moisture and vegetation observations together to improve..."

line 91: WFDEI is defined in section 2 but should be defined here as well or instead of section 2.

lines 97-103 Sentence is much too long. Please break up into separate sentences.

line 115 CTRIP should be defined.

line 119 "detailed hereafter" should be "described in the following sections."

line 122 "They" what is it? Is it the model parameters in the previous sentence? Please be specific.

line 128-129 "net assimilation of CO2" Because the word assimilation is also used in the context of data assimilation, perhaps a different work could be used here? Like "uptake" or "intake"? I just think that using the word assimilation used in the 2 different contexts might confuse the readers.

line 132 "evaporation of"? Or should it read "evaporation due to (i) plant transpiration"?

line 140 What is "it"? Snow scheme or soil diffusion scheme?

line 140 "Richard's " should be "Richards' " (apostrophe after the s) and you need a reference: Richards, L.A., 1931. Capillary conduction of liquids in porous mediums. Physics 1, 318 – 333

line 143 Need a reference for the Brooks and Corey model.

line 187 The LSM is represented by the letter M, but that is not used until eqn. 5. Perhaps better to name M closer to eqn. 5 in the text.

line 237 Should "harmonies" be "harmonious"?

line 297 Is "discharge" "river discharge"? If so please state this.

line 307 "model ability" should be "model's ability"

line 341-345 This is a long sentence and should be broken up. The last bit "...LAI (for SSM and LAI)." doesn't make sense to me, please clarify how LAI is for SSM and LAI? Please make sure that LAI is defined.

line 386 is also unclear with "data set is consistent over time" consistent with what exactly?

Section 3.3 title could be "Impact of the Analysis".

line 390 section 222 should be 2.2.2

Line 400. "Correlation (RMSD)" Please explain what RMSD is it the root mean square deviation, the difference or the sample standard deviation?

line 411 "good values" is vague. Do you mean "high correlation values"?

In the text, the differential terms such as delta (SSM)/ delta (LAI) are missing the superscript that is included in equation 9. Line 450 the lack of superscripts renders that term particularly unhelpful.

line 425 should be "higher than those" not higher compared to"

line 469 Should be "Jacobian's" not "Jacobians"

line 518 Where is "Eff." defined? I would change sentence to "greater than 0 and with 22 gauge stations reporting Eff greater than 0.5."

line 521 Change "superior" to "greater than" or use the mathematical symbol ">" in this paragraph.

line 521 Change to "(3 stations report a decrease > 0.05)"

line 532 Where is "open-loop" defined?

Line 544 MTE needs to be defined.

Line 565 What is an "excessive Jacobian"?

Line 567 What is "They"? and what is the "force-restore version" version of what?

Line 586 which "model" and what is "It" in "It that accounts for the texture-based..."

Line 577 " system too reliant on the chosen forcing" might be better.

Line 573 "they exhibit" what is they?

Line 595 "elaborated methods" doesn't make sense.

Line 601 Again the term from the Jacobian matrix is missing sub or superscripts.

Line 609 Should be "Can better use of" not "Can a better use of"

Line 630 "too large" could be better as "such large"

Line 643 "suggest an added value on vegetation variables" is unclear. how do these variables add value and what exactly is the value added?

Line 652 should be "assimilating retrieved soil moisture"

Line 655 "Tb" needs "b" as a subscript.

Line 666 Better to write "at Meteo-France; it will account for "

Tables:

Table 1. Under "Model" what do DIF and NIT mean?

Figures:

Figure 1. typo "rigth" should be "right"

Figure 2: What does the shaded area represent? Should explain in the caption. Need full stop at end of sentence.

Figure 3: The panels are very small. I think that all panels should be labeled a, b, c etc. and then referred to in the caption by letter. The top 6 panels appear to be for the median R values and the bottom is for a mean RMSD. This is not mentioned in the caption. What times are used in the creation of the median and mean? "Averaged values are reported..." which values are being averaged? In caption state that w_2 is the second layer of soil.

Figure 4: Needs a label for the x axis. N is not defined in the caption but a number is given for N in each panel. The Jacobian elements need to match equation 9. There is a lack of superscript on the LAI variable. What are the solid blue lines in the histogram? Only the lines are defined in caption. Is there a vertical line drawn at 0.0? That should be stated because it is hard to see.

Figure 5: State which column is which and which row is which. "Rows from top to bottom represent averaged analysis increments for all months Feb, May, Aug and Nov from 2000-2012...."

Figure 6: The y axis is not labeled correctly. It should be latitude not 200001-201212. If that is a year and month, it should be in the title or caption. Captial "S" needed. Change to "whole period 2000-2012".

Figure 7: Panels need labels a, b,c and they need to be referenced as such in the panels.

Figure 8: What is N? You don't really need a legend for Red and Green on each of the 6 month plots. Just define in the caption. Panels need labels a, b,c and they need to be referenced as such in the panels. Label the x axis. y-axis is not a probability but a frequency of occurrence. Integral of the Probability function should be equal to one.

Figure 9: Panels need labels a, b,c and they need to be referenced as such in the panels.

Figure 10: In caption, please tell the reader what is Agreste?

[Figure]

Figure 11: Panels need labels a, b,c and they need to be referenced as such in the panels.

Figure 12: The y axis is not labeled correctly. It should be latitude not 200001-201212. The multiple panels are very small and hard to see. I think that you could take the middle row and make it into a new figure. It is not about Evapotranspiration like the top and bottom rows. Please rewrite the second sentence. Be more explicit. For example: Maps of averaged taken over 2000-2012 of a) evapotranspiration...

Figure 13: Rewrite caption. Use full stops. For example: RMSD (a) and correlations (b) between analysed (modelled) ....Panels c and d show Carbon... Panels e and f compare...

Figure 14: Panels need labels a, b,c and they need to be referenced as such in the panels. What is the observation dataset being used? What is the red shaded area? Rewrite: " Monthly RMSD and correlation values between...."

---

## Author Comment (AC1) · 12 Aug 2017

RESPONSE TO REVIEWER #1

*"[...] This paper presents the LDAS-Monde data assimilation system and it evaluate results over Europe. Results are well presented, showing interesting consistent impact of the data assimilation on soil moisture, LAI and hydrological variables such as discharge. It is a innovative study of high interest for the community because it presents multi-variate data assimilation in a coupled land surface and river routing model. The paper is very clearly written and very well presented. I recommend the paper to be published after the minor comments below are accounted for."*

The authors thank anonymous Reviewer 1 for his/her review of the manuscript and for the fruitful comments. Responses to the Reviewer 1 are structured as follow: (1) 1.X: comments from Reviewer 1, (2) Response to 1.X: author's response and author's changes in manuscript when any. For sake of clarity, line and page numbering from the first submission is used.

**1.1 [Page 1, lines 12-13: The first sentence of the abstract should be reformulated to clarify the two objectives of the paper which are testing LDAS-Monde and improving monitoring. It does not make sense to says that LDAS-Monde "is tested ... to increase monitoring accuracy...". Testing itself only allows to evaluate and to asses monitoring accuracy.]**

Response to 1.1
Agreed, the first sentence on the abstract has now been reformulated.
*"In this study, a global Land Data Assimilation system (LDAS-Monde) is tested over Europe and the Mediterranean basin to increase monitoring accuracy for land surface variables."* Is now:
*"In this study, a global Land Data Assimilation system (LDAS-Monde) is applied over Europe and the Mediterranean basin to increase monitoring accuracy for land surface variables."*

**1.2 [Page 2 line 31: Replace "Assimilation impact shows that" by "results show that"]**

Response to 1.2
Agreed, it has now been changed in the revised version of the manuscript.
*Assimilation impact shows that the LDAS works well constraining the model to the observations and that stronger corrections are applied to LAI than to SM.* Is now: "*Results shows that the LDAS works well constraining the model to the observations and that stronger corrections are applied to LAI than to SM.*"

**1.3 [Page 2 line 33: "The assimilation impact's evaluation is succesfully carried out using...". It is not clear what succesfully means here. Is it that it worked technically or that it is a comprehensive evaluation using different data sets, or the results show good performance?. I would just replace by "A comprehensive evaluation of the assimilation impact is conducted using ..."]**

Response to 1.3
Agreed, it has now been changed in the revised version of the manuscript.
*"The assimilation impact's evaluation is successfully carried out using […]"* is now: "*A comprehensive evaluation of the assimilation impact is conducted using […]*"

**1.4 [Page 6, line 147: replace "depth" by "deep"]**

Response to 1.4
Agreed.

**1.5 [Page 8 equation 5: it is not clear what ti and ti+1 are. ti must be the analysis time, but "+1" needs to be clarified: time step? analysis window length?]**

Response to 1.5
It is now clarify in the revised version of the manuscript, it represents the end of the 24-hour assimilation window.
*"The control vector evolution from time $t_i$ to time $t_{i+1}$ is then [...]"* is now *"The control vector evolution from time $t_i$ to the end of the 24-hour assimilation window ($t_{i+1}$) is then [...]"*

**1.6 [Page 8, 219: "WFDEI originates from the ECMWF ERA-Interim reanalysis (Dee et al., 2011) with a spatial resolution of 0.5": replace with by "interpolated to" otherwise it gives the impression that ERA-Interim is at 0.5 degrees resolution which is wrong.]**

Response to 1.6
Agreed, *"WFDEI is based on the ECMWF ERA-Interim reanalysis (Dee et al., 2011) with a spatial resolution of 0.5°, and is corrected [...]"* is now *"WFDEI is based on the ECMWF ERA-Interim reanalysis (Dee et al., 2011) interpolated to a spatial resolution of 0.5°, and is corrected [...]"*

**1.7 [Page 9, line 3: reformulate the sentence to avoid parenthesis (confusing because they do not correspond to mathematical formulation in this case): Where SSMm, SSMo, σm and σo correspond to the model and observation means and standard deviations, respectively.]**

Response to 1.7
Agreed, considered sentence has been replaced by: *"Where $SSM_m$, $SSM_o$, $\sigma_m$ and $\sigma_0$ correspond to the model and observation means and standard deviations, respectively."*

**1.8 [Page 11 line 310: Here the time is a mean time for a given month, whereas earlierin the paper "t" was used for instantaneous time. Replace "time t" by "month mt" or something like that.]**

Response to 1.8
Agreed, *"[...] where $Q_s^t$ is the simulated river discharge (or analysed) at time t and $Q_o^t$ is observed river discharge at time t."* is now: *"[...] where $Q_s^t$ is the simulated river discharge (or analysed) at time t and $Q_o^t$ is observed river discharge at month mt"*. It has also been changed in equation 8.

**1.9 [Page 11 line 310, and page 19 lines 518, 521: replace " Eff. " by "Eff"]**

Response to 1.9
Agreed

**1.10 [Page 13 line 356: "))"]**

Response to 1.10
This typo has now been corrected.

**1.11 [Page 13 lines368-370: "Soil moisture observational and background errors are also scaled by the model soil moisture range, assuming that there is linear relationship between the soil moisture errors and the dynamic range. This was already said lines 354-356. Avoid repetition.]**

Response to 1.11
Agreed, "*[…], assuming that there is linear relationship between the soil moisture errors and the dynamic range*" has now been deleted.

**1.12 [Page 14 table1: "Earth2Observe" by "EartH2Observe" and define NIT.]**

Response to 1.12
Agreed, "*Earth2Observe*" is now "*EartH2Observe*" and NIT has been defined as "*the biomass option selected for the ISBA LSM*" in the caption as well as "Dif" indicating the diffusion scheme of ISBA LSM.
New caption is: "*Summary of the experimental setup used in this study. "Dif" indicates that the diffusion scheme of the ISBA LSM is used, 'NIT' represents the biomass option selected.*"

**1.13 [Page 14 line 377-380: This is not clear. The Jacobian values study could be done using the background forecasts of the analysis experiment. Please clarify.**

Response to 1.13
We agree that the Jacobian values study could be done using the background forecasts of the analysis experiment. In the literature diagnostic studies of the Jacobian values have usually been performed before including (new) observations types (Chevallier and Mahfouf, 2001, Fillion and Mahfouf, 2003, Garand et al., 2001 and Rudiger et al., 2010). In this study it has been decided to follow the same approach as in Rudiger et al., 2010.

We also agree that the sentence pointed out by Reviewer #1 (line 377-380) needs clarifications, it has now been changed in the revised version of the manuscript:

"*Prior to these runs, an analysis experiment without assimilating any observations has also been run over 2000-2012 to study the model sensitivity to the observations through the Jacobians. Studies of the Jacobians values have to be performed before including observations because it is essential to understand the sensitivity of the assimilation system before combining it with observations.*" is now "*Diagnostic studies of the Jacobian values have usually been performed before including new observations types (Chevallier and Mahfouf, 2001, Fillion and Mahfouf, 2003, Garand et al., 2001 and Rudiger et al., 2010). That is why, following Rudiger et al., 2010, an analysis experiment without assimilating any observations has also been run over 2000-2012 to study the model sensitivity to the observations through the Jacobians.*"

The following references have been added to the revised version of the manuscript:

*Chevallier, F., and J.-F. Mahfouf: Evaluation of Jacobians of infrared models for variational assimilation, J. Appl. Meteorol., 40, 1445–1462, doi:10.1175/1520-0450(2001)040, 2001.*

*Fillion, L., and J.-F. Mahfouf: Jacobians of an operational prognostic cloud scheme, Mon. Weather Rev., 131, 2838–2856, doi:10.1175/1520-0493(2003)131, 2003.*

*Garand, L., Turner, D. S., Larocque, M., Bates, J., Boukabara, S., Brunel, P., Chevallier, F., Deblonde, G., Engelen, R., Hollingshead, M., Jackson, D., Jedlovec, G., Joiner, J., Kleespies, T., McKague, D. S. McMillin, L., Moncet, J.-L., Pardo, J. R., Rayer, P. J., Salathe, E., Saunders, R., Scott, N. A., Van Delst, P., Woolf, H.: Radiance and Jacobian intercomparison of radiative transfer models applied to HIRS and AMSU channels, J. Geophys. Res., 106, 24,017–24,031, doi:10.1029/2000JD000184, 2001.*

**1.14 [Page 14 line 390: "section 222" by "section 2.2.2"]**

Response to 1.14
Agreed.

**1.15 [Page 14 392-395 and figure 2: It is not clear how the monthly LAI correlation are computed with one observation every ten days.]**

Response to 1.15
Reviewer #1 highlights a part of the text that needs to be clarified, thanks for pointing this out. First, scores are provided for each year. For LAI (available every 10 days) they are based on a maximum of 36 values for each pixels. Then with figure 2 seasonal scores are provided: values for LAI for a considered month encompass all the 2000-2012 period, i.e. for January we have used values of all January months within 2000-2012 (i.e. a maximum of 13 x 3 LAI values). It is now clarify in the revised version of the manuscript by referring to seasonal scores instead of monthly scores (in the text and in the caption of figure 2).

"***RMSD exhibits however a strong seasonal dependency as illustrated by*** *Error! Reference source not found. **(blue line) with values close to 1 m²m⁻² from June to October***" is now "***Figure 2 (blue line) illustrates seasonal RMSDs (fig. 2a) and correlations (fig. 2b) between LAI from the open-loop and the GEOV1 LAI estimates over 2000-2012. From fig. 2a, a strong seasonal dependency of RMSD is noticeable with values close to 1 m²m⁻² from June to October.***"

Caption of figure 2 is now: "***Seasonal a) RMSD and b) correlation values between leaf area index (LAI) from the open-loop, analysis and GEOV1 LAI estimates from the Copernicus Global Land Service project over 2000-2012.***"

**1.16 [Page 16 lines 426-429: "Sensitivity of LAI to changes in soil moisture (Table 2, bottom rows) is generally weaker than that of SSM (Table 2 top rows) suggesting that although control variables related to soil moisture will be impacted by the assimilation of LAI, they would be even more impacted by the assimilation of SSM." It is not possible to directly compare jacobians values of dSSM/dW (top row) and dLAI/dw (bottom) as they don't have the same unit at all. So, the logic of this sentence does not work. The authors should elaborate the analysis of the table results to come to the conclusion that SSM assimilation should have more impact than LAI assimilation.]**

Response to 1.16
Reviewer #1 is absolutely right, the logic of this sentence doesn't work at all and it is removed from the revised version of the manuscript. Impact of each observation-types, LAI and surface soil moisture on the analysis is currently underway at Météo-France.

"***Sensitivity of LAI to changes in soil moisture (****Error! Reference source not found.****, bottom rows) is generally weaker than that of SSM (****Error! Reference source not found. ****top rows) suggesting that although control variables related to soil moisture will be impacted by the assimilation of LAI, they would be even more impacted by the assimilation of SSM***" is now: "***Sensitivity of LAI to changes in soil moisture (****Error! Reference source not found.****, bottom rows) suggests that control variables related to soil moisture will also be impacted by the assimilation of LAI.***"

**1.18 [Page 17: lines 451-452: just say "January", "June" and "October" to be consistent with the wording used just above (eg line 444) to present this figure. The figure caption makes it clear that it is for the multi-year period.]**

Response to 1.18
Agreed.

**1.19 [Page 18: line 500: "Areas where positive analysis increments were found for LAI (Figure 5) are marked by a decrease in drainage and runoff (in red on Figure 9) while evapotranspiration increases (in blue Figure 9)." It is not that systematic: drainage and runoff impact is more patchy than LAI increments. I would replace "are marked by" by "tend to correspond to"]**

Response to 1.19
Reviewer #1 is right, this statement has to be smoothed: "*are marked by*" is replaced by "*tend to correspond to*".

**1.20 [Page 19, line 516 and Figure 10 caption: "46.0858 N-3.21641E" is it necessary to give 5 digits? It is finer than the model resolution and if the purpose is the traceability of the observed GY location is there an site id number that could be used instead? It would be clearer in the paper to stick to two digits latitude-longitude information (as in line 523 for the discharge).]**

Response to 1.20
Agreed, to our knowledge Agreste does not provide id number. New caption is: *"a) Correlation values for the above ground biomass from the open-loop with grain yields estimates from Agreste French agricultural statistics portal (http://agreste.agriculture.gouv.fr) over 45 sites in France plotted against correlations between the same quantities but above ground biomass from the analysis; b) same as a) for RMSD values; c) scaled anomalies time-series of above ground biomass from the open-loop (black dashed line) the analysis (black solid line) and grain yields observations (red solid) for one site in Allier, France (46.09N-3.21E)."*

**1.21 [Figure 14: not discussed in the paper]**

Response to 1.21
Corrected, it was a typo referring to the wrong figure when discussing GPP evaluation. Also, figure 14 is now Figure 15 as Reviewer #2 suggested to split figure 12 in two figures.

**1.22 [Page 20 line 552-555 and Figure 12: I would expect figure 12 d and l to be identical as they are both indicated to show 'analysis-Model' evapotranspiration. Please clarify.]**

Response to 1.22
Figure 12 has now been split into two figures (as suggested by Reviewer #2), one representing evapotranspiration (using estimates from both GLEAM, top row, and FLUXNET-MTE, bottom row projects) and one for Gross Primary Production (using estimates from the FLUXNET-MTE project). When referring to the labelling of the current version of the manuscript, if figure 12 d) and i) represent the same information, not the same period is considered: 2000-2012 for 12 d) and 2000-2011 for 12 i). It explains that even if geographical patterns are similar color intensity is slightly different.

**1.23 [Page 21: line 569: 'the observation operator is the very thin top layer' this is not correct. Replace by 'the model equivalent was the very thin top layer'. Also replace "that is a thick layer" by "that was a thick layer" to be consistent with the first part of the sentence line 568.]**

Response to 1.23
Agreed, it is now corrected in the revised version of the manuscript.

---

## Author Comment (AC2) · 12 Aug 2017

RESPONSE TO REVIEWER #2

*"In the paper "Sequential assimilation of satellite-derived vegetation and soil moisture products using SURFEX_v8.0: LDAS-Monde assessment over the Euro-Mediterranean area", the authors are motivated "to increase monitoring accuracy" of land surface variables such as soil moisture (SM) and leaf area index (LAI) over the European Mediterranean region. They use a global Land Surface Assimilation system called "LDAS-Monde" to assimilate both SM and LAI in experiments to assess the effects on the model land surface variables. The model results are compared to independent observations of river discharge, land evapo-transpiration and different agricultural statistics and measures. I recommend that this paper undergo major revisions.[…]"*

The authors thank anonymous Reviewer 2 for his/her review of the manuscript and for the fruitful comments. Responses to the Reviewer 2 are structured as follow: (1) 2.X: comments from Reviewer 2, (2) Response to 2.X: author's response and author's changes in manuscript when any. For sake of clarity, line and page numbering from the first submission is used.

**2.1 [The last few sections are not clearly organised or written. The abstract and introduction are very clear about the purpose of the paper, however, the sections from 3.4 onwards lack clarity and do not lead the reader to a direct interpretation of the results as stated in the abstract. The reader would have difficulty coming to the conclusions that are put forward in the abstract. I recommend that these sections be carefully re-written.]**

Response to 2.1
Agreed, section 3.4 on evaluation of the analysis impact as well as the discussion section were carefully re-written, according to Reviewer #2 comment 2.1 but also to comments 2.3 to 2.5, 2.7 and 2.8. Responses to technical corrections (see Responses to comments 2.32 to 2.54) also helped re-writing section 3.4 onwards. Should a revised version of this paper be accepted in GMD, a copy editing work will be performed.

Section 3.4 (L.493-544) is now:

[revised manuscript text omitted]

**2.2 [The title includes acronyms that should be spelled out in full if they really need to be used at all. Not everyone is familiar with the acronym "LDAS" or "SURFEX". Is it necessary to put a specific version number of "SURFEX" in the title?]**

Response to 2.2
Global Model Development journal (GMD) proposes different manuscript types including Model Description Paper where it is a requirement to give the model name and version number (or other unique identifier) in the title, please see:
https://www.geoscientific-model-development.net/about/manuscript_types.html#item5

Although our manuscript has been submitted as a Model Evaluation Paper, and because it is part of the SURFEX special issue we find it useful to indicate the specific version number of the SURFEX modelling platform.

**2.3 [In many places the RMSD and correlations computed are discussed in the same sentence and this creates confusion. It would be simpler to have two or more shorter sentences that are more explicit about which measure is being used for the comparison. I think that overall the authors have chosen brevity over clarity.]**

Response to 2.3
Agreed, the considered sections (mainly sections 3.1 and 3.4) has been revised, please see Response to comments 2.1, 2.5 and 2.32.

**2.4 [Also, the sign (positive or negative) of a change in the metric used is given without explaining what the change means in terms of the variables or the models. A physical interpretation of such results would be helpful.]**

Response to 2.4
Agreed, Authors believe that Reviewer #2 refers to the description of figure 13 (now figure 14) it has been revised, please see also Response to comments 2.1 and 2.5.
It is now: "*Figure 14 shows maps of RMSD (Fig.14a) and correlations (Fig.14b) differences: scores between the analysis and the GLEAM estimates minus scores between the open-loop and the GLEAM estimates. Most of the pixels present negative values for differences in RMSD (76% fig.14 a) indicating that for those pixels RMSDs from the analysis are smaller than those from the open-loop. Most of the pixels present positive values differences in correlations (80% fig.14 b) indicating that for those pixels correlations from the analysis are higher than those from the open-loop. It shows the added value of the analysis when compared to the open-loop. Evapotranspiration from the open-loop and analysis has also been evaluated using FLUXNET-MTE estimates of evapotranspiration (2000-2011). Results are illustrated by Figure 12e to h and Figure 14e and f. They are similar of those obtained using GLEAM estimates: over the whole domain most of the pixels present negative values for differences in RMSD (70%), most of the pixels present positive values for differences in correlation (79%).*"

**2.5 [Please rewrite the sentence in 535-536 "Most of the differences in RMSD are negative..." RMSD is a strictly positive or zero value. Are the differences in RMSD between two different data sets being compared? Could the authors please write two sentences that explain this point more explicitly? It appears to be an important point as it is going to "show the added value of the analysis".]**

Response to 2.5
Agreed, the whole paragraph has been revised for a better understanding. It is now: "*However the comparison with the GLEAM satellite-derived estimates is rather positive, as illustrated in Error! Reference source not found. representing evapotranspiration from the open-loop (Fig.12a) , GLEAM estimates (Fig.12b), the analysis (Fig.12c) and their differences (Fig.12d). Open-loop simulation of evapotranspiration tends to over-estimate the GLEAM product over most of Europe, particularly over France and the Iberian Peninsula, North Africa. Analysis is able to reduce this bias (Error! Reference source not found.d). Figure 14 shows maps of RMSD (Fig.14a) and correlations (Fig.14b) differences: scores between the analysis and the GLEAM estimates minus scores between the open-loop and the GLEAM estimates. Most of the pixels present negative values for differences in RMSD (76% fig.14 a) indicating that for those pixels RMSDs from the analysis are smaller than those from the open-loop. Most of the pixels present positive values for differences in correlations (80% fig.14 b) indicating that for those pixels correlations*

*from the analysis are higher than those from the open-loop. It shows the added value of the analysis when compared to the open-loop.*"

**2.6 [The figures (details are given in the technical comments below) need work as well. For example, in Figure 8: What is N? You don't really need a legend for the red and green lines on each of the 6 month plots. Just define this in the caption. Panels need labels a, b,c and they need to be referenced as such in the caption. Please label the x axis with variable name and units. Most importantly, the y-axis is not a probability but a frequency of occurrence (the caption even says "histogram" which is correct). The integral of the probability distribution function should be equal to one by definition.]**

**Response to 2.6**
Agreed, figures have been improved accordingly. Please see also Responses to technical comments 2.56 to 2.68. Regarding the y-axis of figure 8, it should be labelled 'Probability density', it represents the counts normalized to form a probability density, i.e., the area (or integral) under the histogram will sum to 1. This is achieved dividing the count by the number of observations times the bin width and not dividing by the total number of observations. Y-labels on figures 4, 8 and 11 (b, c and d) are now 'Probability density'.

**2.7 [The last two paragraphs of section 3.4 are very unclear. The sentence "From Figures 13 d and e and Figure 13 one may notice...seasonal decrease and increase of the scores." does not make sense. Perhaps the authors need several sentences here that are more precise about which figures support which conclusions. Again, the RMSD and correlation should be discussed separately to make the evidence clearer. Line 555 contains another confusing sentence. "differences in RMSD and correlations are negative and positive: 70% and 79%. This just doesn't make any sense. Are there percent changes in a particular direction? If so, what are the implications for the model output or the physical system?]**

Response to 2.7
Agreed, the considered section (section 3.4) has been revised, please see Response to comments 2.1 and 2.5 as well.

**2.8 [Section 4.1 is called "Can different data assimilation techniques improve the analysis?" I don't believe that this particular question has been answered in this section by the work presented here. I think that alternative methods are proposed and discussed but the actual results in the paper do not answer the question whether one techniques is better than another. If the section could be renamed, that would be more clear.]**

Response to 2.8

Agreed, section 4.1 has been renamed, it is now: "*Towards different data assimilation techniques to improve the analysis*"

**2.9 [Abstract: SM is defined but later in the main body SSM is used. Perhaps should just define SSM in abstract and stick to it?]**

Response to 2.9
Agreed, SSM is now defined from the abstract.

**2.10 [Introduction: Define acronyms MODCOU and SAFRAN]**

Response to 2.10

Agreed, both acronyms are in French, MODCOU stands for: "MODèle COUplé" , SAFRAN stands for: "Système d'Analyse Fournissant des Renseignements Atmosphériques à la Neige".

**2.11 [Section 3.1 Should be "Consistency between the model and observations"]**

Response to 2.11
Agreed.

**2.12 [Need to state what is consistent with what. "Observations consistency over time…" is not clear. Do you mean that one set of observations are consistent with another observation set? or with the model output?]**

Response to 2.12
Those dataset have been evaluated/compared against long term in situ measurements/re-analysis of soil moisture and LAI with no decrease of quality over time. That is what authors mean by '*consistency over time*'. For instance, Albergel et al., 2013a have compared the ESA CCI soil moisture product against the ERA-Interim/Land re-analysis over 1979-2010 for all the 3-yr periods within 1979-2010. Correlations values were found rather stable with a small increase over time. Also time-series do not present any spurious jumps or drifts.

In the context of our evaluation and for sake of clarity, it is now emphasise that the data-set consistency against the open-loop is evaluated: "*Observations consistency over time is crucial when assimilating long-term datasets. Several authors assessed the consistency of the ESA CCI soil moisture product with respect to re-analysis products (e.g., Loew et. al., 2013; Albergel et. al., 2013a; 2013b) and in-situ measurements (Dorigo et. al., 2015, 2017). […] To verify the results from literature for the spatial and temporal domain considered in this study a consistency evaluation both for SSM and LAI has been performed*" Is now: "*Consistency over time is crucial when assimilating long-term datasets. Several authors assessed the consistency of the ESA CCI soil moisture product with respect to re-analysis products (e.g., Loew et. al., 2013; Albergel et. al., 2013a; 2013b) and in-situ measurements (Dorigo et. al., 2015, 2017). […] To verify the results from literature for the spatial and temporal domain considered in this study a consistency evaluation both for SSM and LAI against the open-loop experiment has been performed*".

**2.13 [lines 61-63: should read "perform best for plant productivity...to used soil moisture and vegetation observations together to improve…"]**

Response to 2.13
Agreed.

**2.14 [line 91: WFDEI is defined in section 2 but should be defined here as well or instead of section 2.]**

Response to 2.14
Agreed, it is now defined in the introduction only.

**2.15 [lines 97-103 Sentence is much too long. Please break up into separate sentences.]**

Response to 2.15
Agreed, it is now two sentences: "**Section 2 presents the LDAS-Monde system, i.e. (i) the $CO_2$ responsive version of the ISBA LSM and the soil diffusion scheme, (ii) the CTRIP hydrological model and its coupling with ISBA, (iii) the atmospheric forcing used to drive the system, (iv) the equations of the SEKF and (v) the assimilated remotely sensed observations**

**dataset as well as the datasets used to assess the analysis impact. The latter is evaluated using agricultural statistics over France, river discharge, satellite-derived estimates of land transpiration and spatially gridded estimates of up-scaled gross primary production from the FLUXNET network.**"

**2.16 [line 115 CTRIP should be defined.]**

Response to 2.16
CTRIP has already been defined in the introduction l.94-96: "*Having a daily interactive coupling between ISBA and the CNRM (Centre National de Recherches Météorologiques) version of the TRIP (Total Runoff Integrating Pathways, Oki et al., 1998) river routing model (CTRIP hereafter)*".

**2.17 [line 119 "detailed hereafter" should be "described in the following sections."]**

Response to 2.17
Agreed.

**2.18 [line 122 "They" what is it? Is it the model parameters in the previous sentence? Please be specific.]**

Response to 2.18
Agreed, "*ISBA models the basic land surface physics requiring only a small number of model parameters. They depend on the soil and vegetation types.*" Is now: "*ISBA models the basic land surface physics requiring only a small number of model parameters. The latter ones depend on the soil and vegetation types.*"

**2.19 [line 128-129 "net assimilation of CO2" Because the word assimilation is also used in the context of data assimilation, perhaps a different work could be used here? Like "uptake" or "intake"? I just think that using the word assimilation used in the 2 different contexts might confuse the readers.]**

Response to 2.19
Agreed, "*net assimilation of $CO_2$*" has been replaced by "*$CO_2$ uptake*".

**2.20 [line 132 "evaporation of"? Or should it read "evaporation due to (i) plant transpiration"?]**

Response to 2.20
Agreed, "*evaporation of*" is now "*evaporation due to*".

**2.21 [line 140 What is "it"? Snow scheme or soil diffusion scheme?]**

Response to 2.21
It is now clarify: "*The multi-layer soil diffusion scheme version is based on the mixed form of the Richards' equation (Richards, 1931) and explicitly solves the one-dimensional Fourier law. Additionally, it incorporates soil freezing processes developed by Boone et al. (2000) and Decharme et al. (2013).*" is now "*The multi-layer soil diffusion scheme version (ISBA-Dif) is based on the mixed form of the Richards' equation (Richards, 1931) and explicitly solves the one-dimensional Fourier law. Additionally, ISBA-Dif incorporates soil freezing processes developed by Boone et al. (2000) and Decharme et al. (2013).*"

**2.22 [line 140 "Richard's " should be "Richards' " (apostrophe after the s) and you need a reference: Richards, L.A., 1931. Capillary conduction of liquids in porous mediums. Physics 1, 318 – 333]**

Response to 2.22
Agreed, reference to Richards, 1931 has now been added to the manuscript.

**2.23 [line 143 Need a reference for the Brooks and Corey model.]**

Response to 2.23
Agreed, the following reference has now been added to the revised version of the manuscript:
***Brooks, R. H., and A. T. Corey: Properties of porous media affecting fluid flow, J. Irrig. Drain. Div. Am. Soc. Civ. Eng., 17, 187–208, 1966.***

**2.24 [line 187 The LSM is represented by the letter M, but that is not used until eqn. 5. Perhaps better to name M closer to eqn. 5 in the text.]**

Response to 2.24
Authors prefer not to change this sentence as it is important to indicate at this stage that $x$ is the control vector that represents the prognostic equations of the LSM $M$. If $M$ is also mentioned close to Equation 5, it will be redundant.

**2.25 [line 237 Should "harmonies" be "harmonious"?]**

Response to 2.25
Authors thanks Reviewer #2 for pointing out this typo, it is now corrected.

**2.26 [line 297 Is "discharge" "river discharge"? If so please state this.]**

Response to 2.26
Agreed, it is now corrected.

**2.27 [line 307 "model ability" should be "model's ability"]**

Response to 2.27
Agreed, it is now corrected.

**2.28 [line 341-345 This is a long sentence and should be broken up. The last bit "...LAI (for SSM and LAI)." doesn't make sense to me, please clarify how LAI is for SSM and LAI? Please make sure that LAI is defined.]**

Response to 2.28
The considered sentence is now reduced and clarified, "***The LDAS used in this study is designed as follow; $x$ is the 8-dimensional control vector including soil layers 2 to 8 (representing a depth from 1 cm of 100cm) and LAI propagated by ISBA LSM. $y_o$ is the 2-dimensional observation vector (SSM, LAI) and the model counterparts of the observations are the second layer of soil of ISBA LSM ($w_2$ between 1 and 4 cm) and LAI (for SSM and LAI).***" is now: "***The LDAS used in this study is designed as follow; $x$ is the 8-dimensional control vector including soil layers 2 to 8 (representing a depth from 1 cm to 100cm) and LAI propagated by ISBA LSM. $y_o$ is the 2-dimensional observation vector (SSM, LAI). The model counterparts of the observations are the second layer of soil of ISBA LSM ($w_2$ between 1 and 4 cm) and LAI for SSM and LAI observations, respectively.***"

**2.29 [line 386 is also unclear with "data set is consistent over time" consistent with what exactly?**

Response to 2.29
Please see Response to 2.12.

**2.30 [Section 3.3 title could be "Impact of the Analysis".]**

Response to 2.30
Agreed.

**2.31 [line 390 section 222 should be 2.2.2]**

Response to 2.31
Agreed.

**2.32 [Line 400. "Correlation (RMSD)" Please explain what RMSD is it the root mean square deviation, the difference or the sample standard deviation?]**

Response to 2.32
It is now clarify in the text. "*Over the same period, correlation (RMSD) between GEOV1 LAI and […]*" is now "*Over the same period, correlation and Root Mean Square Differences (RMSD) between GEOV1 LAI and SURFEX-CTRIP LAI estimates is 0.75 and 0.85 $m^2m^{-2}$*"

**2.33 [line 411 "good values" is vague. Do you mean "high correlation values"?]**

Response to 2.33
Agreed, "*Low correlations values are found in desert areas (over the Sahara), high elevation (e.g. over the Alps) and at high latitudes whereas good values […]*" is now "*Low correlations values are found in desert areas (over the Sahara), high elevation (e.g. over the Alps) and at high latitudes whereas high correlations values […]*"

**2.34 [In the text, the differential terms such as delta (SSM)/ delta (LAI) are missing the superscript that is included in equation 9. Line 450 the lack of superscripts renders that term particularly unhelpful.]**

Response to 2.34
Agreed, it is now corrected in Table 2 and through the whole manuscript.

**2.35 [line 425 should be "higher than those" not higher compared to"]**

Response to 2.35
Agreed.

**2.36 [line 469 Should be "Jacobian's" not "Jacobians"]**

Response to 2.36
Agreed.

**2.37 [line 518 Where is "Eff." defined? I would change sentence to "greater than 0 and with 22 gauge stations reporting Eff greater than 0.5."]**

Response to 2.37

It is defined in section "2.2.4 Evaluation data sets and strategies": "*Impact on Q is evaluated using correlation, RMSD as well as the efficiency score (Eff) (Nash and Sutcliff, 1970). Eff evaluates the model's ability to represent the monthly discharge dynamics and is given by:*

$$Eff = 1 - \frac{\sum_{mt=1}^{T}(Q_s^{mt} - Q_o^{mt})^2}{\sum_{mt=}^{T}\left(Q_o^{mt} - \overline{Q_o^{mt}}\right)^2} \qquad (8)$$

*where $Q_s^t$ is the simulated river discharge (or analysed) at time t and $Q_o^t$ is observed river discharge at month mt. The Eff can vary between $-\infty$ and 1. A value of 1 corresponds to identical model predictions and observed data. A value of 0 implies that the model predictions have the same accuracy as the the mean of the observed data. Negative values indicate that the observed mean is a more accurate predictor than the model simulation.*"

"*Over 2000-2010, 48 of 83 gauge station present Eff values greater than 0, 22 greater than 0.5*" is now "*Over 2000-2010, 48 of 83 gauge station present Eff values greater than 0 and 22 gauge stations report Eff greater than 0.5*"

**2.38 [line 521 Change "superior" to "greater than" or use the mathematical symbol ">" in this paragraph.]**

Response to 2.38
Agreed, "*superior*" is now "*greater than*".

**2.39 [line 521 Change to "(3 stations report a decrease > 0.05)"]**

Response to 2.39
Agreed, "*(3 present a decrease superior to 0.05)*" is now "*(3 stations report a decrease greater than 0.05)*"

**2.40 [line 532 Where is "open-loop" defined?]**

Response to 2.40
It is defined in the introduction, L.80 "*However, the assimilation was not successful in improving the representation of river discharge within MODCOU compared to an open-loop (i.e. no assimilation) simulation.*"
For sake of clarity, it now is repeated in section 2.3 on experimental setup: "*SURFEX-CTRIP was spun up by cycling twenty times through the year 1990, then a 10-yr model run is allowed before considering both an open-loop (a model run with no assimilation) and an analysis experiment over 2000-2012.*"

**2.41 [Line 544 MTE needs to be defined.]**

Response to 2.41
MTE is defined L.322-323, section 2.2.4 on Evaluation data sets and strategy: "*The up-scaled FLUXNET GPP and evapotranspiration were derived from the FLUXNET network using a model tree ensemble (FLUXNET-MTE hereafter) approach as described in Jung et al. (2009).*"

**2.42 [Line 565 What is an "excessive Jacobian"?]**

Response to 2.42

Wording is indeed not clear, by 'excessive' Authors meant 'outliers'. It is now corrected (please see also Response to 2.43).

**2.43 [Line 567 What is "They"? and what is the "force-restore version" version of what?]**

Response to 2.43
For sake of clarity, "*They were however obtained using the force-restore version with three layers of soil.*" is now "*Those outliers in the Jacobian's values were however obtained using the force-restore version of the ISBA LSM with three layers of soil and not with the diffusion soil scheme: ISBA-Dif.*"

**2.44 [Line 586 which "model" and what is "It" in "It that accounts for the texture-based..."]**

Response to 2.44
For sake of clarity, "*Soil moisture observations and background errors were scaled using the model dynamic range. It accounts for texture-based spatial variability in the error and assumes that the soil moisture errors and the dynamic range have a linear relationship.*" is now "*Soil moisture observations and background errors were scaled using the open-loop soil moisture dynamical range. The scaling accounts for texture-based spatial variability in the error and assumes that the soil moisture errors and the dynamic range have a linear relationship.*"

**2.45 [Line 577 " system too reliant on the chosen forcing" might be better.]**

Response to 2.45
Agreed, "*The SEKF is also limited in correcting errors from the atmospheric forcing uncertainty making the system relying too much on the chosen forcing.*" Is now "*The SEKF is also limited in correcting errors from the atmospheric forcing uncertainty making the system too reliant on the chosen forcing.*"

**2.46 [Line 573 "they exhibit" what is they?]**

Response to 2.46
For sake of clarity, "*they*" is now "$\frac{\partial SSM^t}{\partial w_{2-RZ}^0}$*Jacobians*"

**2.47 [Line 595 "elaborated methods" doesn't make sense.]**

Response to 2.47
Agreed, "*elaborated*" is replaced by "*statistical*"

**2.48 [Line 601 Again the term from the Jacobian matrix is missing sub or superscripts.]**

Response to 2.48
Corrected here and through the whole manuscript.

**2.49 [Line 609 Should be "Can better use of" not "Can a better use of"]**

Response to 2.49
Agreed.

**2.50 [Line 630 "too large" could be better as "such large"]**

Response to 2.50

Agreed, "*too large*" is now "*such large*"

**2.51 [Line 643 "suggest an added value on vegetation variables" is unclear. how do these variables add value and what exactly is the value added?]**

Response to 2.51
For sake of clarity, "*Preliminary results from assimilating disaggregated LAI time series and using new LAI minimum values (not shown) suggest an added value on vegetation variables like above-ground biomass and on the representation of river discharge.*" is now "*Preliminary results from assimilating disaggregated LAI time series and using new LAI minimum values (not shown) suggest better representation of vegetation variables like LAI and above-ground biomass as well as an enhanced representation of river discharge compared to an open-loop simulation using the former LAI minimum values.*"

**2.52 [Line 652 should be "assimilating retrieved soil moisture"]**

Response to 2.52
Agreed, it is now corrected in the revised version of the manuscript: "*Despite the proven record of assimilating soil moisture retrieval from […]*" is now "*Despite the proven record of assimilating retrieved soil moisture from […]*".

**2.53 [Line 655 "Tb" needs "b" as a subscript.]**

Response to 2.53
Agreed.

**2.54 [Line 666 Better to write "at Meteo-France; it will account for "]**

Response to 2.54
Agreed.

**2.55 [Table 1. Under "Model" what do DIF and NIT mean?]**

Response to 2.55
It is now clarify in the caption of Table 1, new caption is "*Summary of the experimental setup used in this study. "Dif" indicates that the diffusion scheme of the ISBA LSM is used, 'NIT' represents the biomass option selected.*"

2.56 [Figure 1. typo "rigth" should be "right"]

Response to 2.56
Agreed, new caption is: "*Averaged (left) surface soil moisture from the Climate Change Initiative project of ESA (right) GEOV1 Leaf Area Index from the Copernicus Global Land Service project (for pixels covered by more than 90% of vegetation) over 2000-2012.*"

**2.57 [Figure 2: What does the shaded area represent? Should explain in the caption. Need full stop at end of sentence.]**

Response to 2.57
The shaded area highlights the analysis impact for each considered metric: if the analysis is better than the open-loop then the area between the two lines (red and blue) is shaded in red and if the open-loop is better than the analysis then it is shaded in blue. However as the analysis is

systematically better than the open-loop there is no need to keep it and it is now remove from figure 2 (as well as from figure 14 now figure 15). Panels were also labelled and new caption is: "*Seasonal a) RMSD and b) correlation values between leaf area index (LAI) from the open-loop, analysis and GEOV1 LAI estimates from the Copernicus Global Land Service project over 2000-2012.*" New figure 2 is presented below.

[Figure]

**2.58 [Figure 3: The panels are very small. I think that all panels should be labeled a, b, c etc. and then referred to in the caption by letter. The top 6 panels appear to be for the median R values and the bottom is for a mean RMSD. This is not mentioned in the caption. What times are used in the creation of the median and mean? "Averaged values are reported..." which values are being averaged? In caption state that w_2 is the second layer of soil.]**

Response to 2.58

Agreed, all panels are now labelled and referred to in the caption which is now : "*top row, yearly averaged correlations between satellite-derived surface soil moisture from the Climate Change Initiative project from ESA and the second layer of soil of SURFEX-CTRIP (w₂: 1 cm-4 cm depth) for a) 2000, b) 2006 and c) 2012. d), e) and f) yearly averaged correlation between the GEOV1 leaf area index from the Copernicus Global Land Service project and SURFEX-CTRIP for 2000, 2006 and 2012, respectively. g), h) and i) same as d), e) and f) for RMSD.*" New figure 3 is presented below.

[Figure]

**2.59 [Figure 4: Needs a label for the x axis. N is not defined in the caption but a number is given for N in each panel. The Jacobian elements need to match equation 9. There is a lack of superscript on the LAI variable. What are the solid blue lines in the histogram? Only the lines are defined in caption. Is there a vertical line drawn at 0.0? That should be stated because it is hard to see.]**

Response to 2.59

Indeed information on N is missing, it represents the sampling. For sake of clarity on figure 4, the 90% confidence interval was chosen to define the upper and lower values to exclude outliers on the histograms. In agreement with Reviewer #2 comment, y-label is now 'Probability density', x-label is now 'Jacobians' and the Jacobian elements match equation 9 and all panels are labelled. New caption is: "*Jacobian values distribution: a) to f),* $\frac{\partial SS^{t}}{\partial w_2{}^0}$*(red line),* $\frac{\partial SSM^{t}}{\partial w_4{}^0}$*(cyan line) and* $\frac{\partial SSM^{t}}{\partial w_8{}^0}$*(blue line) all months of January, March, June, August, October and December over 2000-2012, g) to i),* $\frac{\partial LAI^{t}}{\partial LAI^0}$*(red line),* $\frac{\partial LAI^{t}}{\partial w_4{}^0}$*(cyan line) and* $\frac{\partial LAI^{t}}{\partial w_8{}^0}$*(blue line) for all months of January, June and October over 2000-2012. Black solid line represents a value of 0.*" New figure 4 is presented below.

[Figure]

**2.60 [Figure 5: State which column is which and which row is which. "Rows from top to bottom represent averaged analysis increments for all months Feb, May, Aug and Nov from 2000-2012...."]**

Response to 2.60
Agreed, new caption is: "*Rows from top to bottom represent averaged analysis increments for all months of February, May, August and November over 2000-2012. From left to right for 4 control variables are illustrated, leaf area index and soil moisture in the second ($w_2$, 1 cm- 4 cm), fourth ($w_4$, 10 cm-20 cm) and sixth ($w_6$, 40 cm – 60 cm) layer of soil, respectively.*"

**2.61 [Figure 6: The y axis is not labeled correctly. It should be latitude not 200001-201212. If that is a year and month, it should be in the title or caption. Captial "S" needed. Change to "whole period 2000-2012".]**

Response to 2.61
Agreed, 200001-201212 is now removed from the figure and the new caption is: "*Averaged analysis increments for the whole period 2000-2012. Four control variables are illustrated: a) leaf area index and soil moisture in a) the second ($w_2$, 1 cm- 4 cm), b) fourth ($w_4$, 10 cm-20 cm) and c) sixth ($w_6$, 40 cm – 60 cm) layer of soil.*" New figure 6 is presented below.

[Figure]

**2.62 [Figure 7: Panels need labels a, b,c and they need to be referenced as such in the panels.]**

Response to 2.62

Agreed, labels are now reported and all panels and captions has been changed accordingly: "***RMSD maps between leaf area index from the open-loop (analysis) and that from the Copernicus Global Land Service project (GEOV1 index) for a(b) January, e(f) April, c(d) July and e(f) October over 2000-2012.***" New figure 7 is presented below.

[Figure]

**2.63 [Figure 8: What is N? You don't really need a legend for Red and Green on each of the 6 month plots. Just define in the caption. Panels need labels a, b,c and they need to be referenced as such in the panels. Label the x axis. y-axis is not a probability but a frequency of occurrence. Integral of the Probability function should be equal to one.]**

Response to 2.63

Information on N is indeed missing, it represents the sampling which is reported on each month plots. In agreement with Reviewer #2 comment legend for Red and Green are only reported on the first plot, panels are labelled a, b, c...etc, x-axis is now 'Innovations or Residuals' and y-axis is 'Probability density' (please see also response to comment 2.6). New caption is: "***Probability density function of innovation (observations-open-loop in red) and residuals (observations – analysis, in green) for Leaf Area Index for a) February, b) April, c) June, d) August, e) October and f) December over 2000-2012. Sampling (N) is reported on each panel***". New figure 8 is presented below.

[Figure]

**2.64 [Figure 9: Panels need labels a, b,c and they need to be referenced as such in the panels.]**

Response to 2.64

Agreed, panels are now labeled and the new caption is: "*Averaged analysis impact on land surface variables that are indirectly affected over the period 2000-2012: a) drainage, b) runoff, c) evapotranspiration and d) river discharge.*" New figure 9 is presented below.

[Figure]

**2.65 [Figure 10: In caption, please tell the reader what is Agreste?]**

Response to 2.65
Agreed, new caption is: "*a); Correlation values for the above ground biomass from the open-loop with grain yields estimates from Agreste French agricultural statistics portal (http://agreste.agriculture.gouv.fr) over 45 sites in France plotted against correlations between the same quantities but above ground biomass from the analysis; b) same as a) for RMSD values; c) scaled anomalies time-series of above ground biomass from the open-loop (black dashed line) the analysis (black solid line) and grain yields observations (red solid) for one site in Allier, France (46.09N-3.21E).*"

**2.66 [Figure 11: Panels need labels a, b,c and they need to be referenced as such in the panels.]**

Response to 2.66
Agreed, new labels a, b, c and d are reported on panels, also y-axis is now 'Probability density' for panels b to d (please see also Response to comment 2.6). New figure 11 is presented below.

[Figure]

**2.67 [Figure 12: The y axis is not labeled correctly. It should be latitude not 200001-201212. The multiple panels are very small and hard to see. I think that you could take the middle row and make it into a new figure. It is not about Evapotranspiration like the top and bottom rows. Please rewrite the second sentence. Be more explicit. For example: Maps of averaged taken over 2000-2012 of a) evapotranspiration…]**

Response to 2.67
Agreed, y-label is removed and as suggested by Reviewer #2 this figure is now split into 2 figures, one for Evapotranspiration (new figure 12) and on for Gross Primary Production (new figure 13). It also makes them more visible for Readers. New captions and figures are:

*"Figure 12: Top row: maps of averaged evapotranspiration taken over 2000-2012 from a) the model (i.e open-loop), b) the GLEAM estimates, c) the analysis and d) differences between the analysis and model. Bottom row: maps of averaged evapotranspiration taken over 2000-2011 from a) the model (i.e open-loop), b) FLUXNET-MTE estimates, c) the analysis and d) differences between the analysis and model."*

[Figure]

*Figure 13: Maps of averaged Gross Primary Production taken over 2000-2011 from a) the model (i.e open-loop), b) FLUXNET-MTE estimates, c) the analysis and d) differences between the analysis and model."*

[Figure]

**2.68 [Figure 13: Rewrite caption. Use full stops. For example: RMSD (a) and correlations (b) between analysed (modelled) ....Panels c and d show Carbon... Panels e and f compare...]**

Response to 2.68
Agreed, also figure 13 is now figure 14, the new caption is:

*RMSD (a) and Correlations (b) differences between analysed (modelled) evapotranspiration and GLEAM estimates over 2000-2012. c) and d) are similar to a) and b) for Carbon mass flux out of the atmosphere due to Gross Primary Production (GPP) from the analysis (model), and FLUXNET-MTE GPP estimates over 2000-2011. Finally e) and f) are similar to a) and b) for analysed (modelled) evapotranspiration and FLUXNET-MTE evapotranspiration estimates over 2000-2011.*

**2.68 [Figure 14: Panels need labels a, b,c and they need to be referenced as such in the panels. What is the observation dataset being used? What is the red shaded area? Rewrite: " Monthly RMSD and correlation values between...."]**

Response to 2.69
Agreed, panels are now labelled and the red shaded areas has been removed. Figure 14 is now figure 15. The new caption is : "*Seasonal a) RMSD and b) correlation values between the Carbon*

***mass flux out of the atmosphere due to Gross Primary Production on land (GPP) from the open-loop, analysis and FLUXNET-MTE estimates over 2000-2011.***" New figure 15 is presented below.

[Figure]